# POSTERIOR PROBABILITY-BASED LABEL RECOVERY ATTACK IN FEDERATED LEARNING

## ABSTRACT

Recent works have proposed analytical attacks that can recover batch labels from gradients of a classification model in *Federated Learning* (FL). However, these studies do not explain the essence of label leakage or show the scalability of other classification variants. In this paper, we demonstrate the root cause of label leakage from gradients and propose a generalized label recovery attack by estimating the posterior probabilities. Beginning with the *focal loss* function, we derive the relationship among the gradients, labels and posterior probabilities in a concise form. Then, we explain the essential reasons for such findings from the perspective of the *exponential family*. Furthermore, we empirically observe that positive (negative) samples of a class have approximate probability distributions. This key insight enables us to estimate the posterior probabilities of the target batch from an auxiliary dataset. Integrating the above elements, we finally present our label attack that can directly recover the batch labels of each class in realistic FL settings. Evaluation results show that on an untrained model, our attack can achieve over 96% *Class-level label Accuracy (ClsAcc)* and 95% *Instance-level label Accuracy (InsAcc)* on different groups of datasets, models and activations. For a training model, our approach reaches more than 90% InsAcc on different batch sizes, class imbalance ratios, temperature parameters or label smoothing factors.

## 1 INTRODUCTION

Federated Learning (FL) has become a popular paradigm for training machine learning models in privacy-sensitive applications (McMahan et al., 2017; Yang et al., 2019). In FL, the clients compute gradients on their local devices and then send the gradients to the server for aggregation and global model update. Since the private data is preserved on the client side, FL is supposed to offer more privacy protection than the centralized learning paradigm. However, a category of attacks called *gradient inversion* has shown that the shared gradients can be exploited to reconstruct the training data (Zhu et al., 2019; Geiping et al., 2020; Yin et al., 2021). Moreover, some analytical attacks can recover the labels from gradients of a classification model by analyzing the relationship between the gradients and the labels (Zhao et al., 2020; Wainakh et al., 2022; Ma et al., 2023). However, none of these works explain the nature of label recovery or exhibit the applicability to other classification problems. We thus raise the following key questions: (i) *what is the essence of label leakage from gradients?* and (ii) *how to implement a generalized attack for label recovery?*

In this paper, we explore and answer these two questions from both theoretical and practical perspectives. In particular, starting from the *focal loss* function, we first derive an important relationship among the gradients, labels and posterior probabilities in a concise form. This conclusion can be applied to a variety of loss functions, which reveals the connection between the gradients and the labels in a classification model. Then we explain the fundamental reason for our findings from the *exponential family* of distributions. We show that the gradient w.r.t. logits is the expectation of the target labels, which provides a convenient way to reduce computation costs but opens a backdoor for label leakage. Finally, we propose a generalized attack for label recovery by estimating the posterior probabilities of the target batch from an auxiliary dataset. The key insight is based on our empirical observation that the positive (negative) samples of a class have approximate probability distributions. By fitting the auxiliary dataset into the global model, we can estimate the target posterior probabilities, and then recover the labels of a specified class by substituting the gradients and the posterior probabilities into the derived formula.

Our main contributions are summarized as follows:

- For the first time, we investigate the root cause of label leakage from gradients, and find the gradient w.r.t. logits is only related to the posterior probabilities and the target labels in various loss functions, such as focal loss and binary cross-entropy loss.

- We explain the intrinsic reason for our findings from the perspective of the exponential family, and conclude that the combination of cross-entropy loss and Softmax (or Sigmoid) activation function opens a backdoor for the label restoration attacks.

- We evaluate our attack on a variety of FL settings and classification variants, and demonstrate that it outperforms the prior attacks in terms of *Class-level label Accuracy (ClsAcc)* and *Instance-level label Accuracy (InsAcc)*.

## 2 RELATED WORK

Now we review some of the works most related to ours.

**Federated Learning.** Federated Learning (FL) is a privacy-preserving machine learning paradigm that enables multiple clients to collaboratively train a global shared model without collecting their private data (McMahan et al., 2017; Yang et al., 2019). In FL, the clients train the shared model locally and then send the model update (i.e., the gradient of the model parameters) to the server. The server aggregates the uploaded gradients from the selected clients and updates the global model. The training procedure is repeated until the global model converges. Since the private data is preserved on the client side, FL is widely used in privacy-sensitive applications such as finance (Long et al., 2020; Shingi, 2020) and healthcare (Xu et al., 2021; Nguyen et al., 2022).

**Gradient Inversion Attacks.** Zhu et al. (2019) propose the first gradient inversion attack, which can reconstruct the training data $x$ and corresponding labels $y$ from the shared gradients in FL. An honest-but-curious attacker generates a batch-averaged dummy gradient by fitting the global model with a batch of dummy data, and then iteratively updates the dummy data to minimize the distance between the dummy gradient and the target gradient. Geiping et al. (2020) use cosine similarity as the error function and add total variation as a regularization term to improve the quality of reconstruction. Yin et al. (2021) present group consistency regularization to enhance the restoration of the object locations in the images. (Jeon et al., 2021) exploit a generative model pre-trained on the data distribution to improve the quality of the recovered images. Moreover, recent studies also propose gradient inversion attacks in other tasks, such as natural language processing (Gupta et al., 2022; Balunovic et al., 2022) and speech recognition (Dang et al., 2022). The success of these attacks is based on an underlying assumption that the gradient is an approximate bijection of the data. Thus, decreasing the gap between gradients equals optimizing the dummy data towards the real data.

**Analytical Label Attacks.** Zhao et al. (2020) propose an analytical label attack named iDLG, which can directly infer the label from $\nabla W$ of the classification layer. They derive that the gradient w.r.t the logit $z_j$ equals to $\sigma(z_j) - 1$ if $j$ is the target index $c$ of the one-hot label, and $\sigma(z_j)$ if $j \neq c$, where $\sigma(\cdot)$ denotes the Softmax function. When the model uses a non-negative activation, such as ReLU or Sigmoid, $\nabla W_c$ consists of negative values, while the other rows are positive. Thus, the attacker can extract the label by simply comparing the signs of the gradient $\nabla W$. However, iDLG only applies to single-batch labels, and the activation of the model must be non-negative. Wainakh et al. (2022) exploit the direction and magnitude of $\nabla W$ to determine how many instances of each class are in the target batch. They formulate the problem as $\sum_{i=1}^{M} \nabla W_j = \lambda_j m + s_j$, where $\lambda_j$ is the number of batch labels of the $j$th class, $m$ is the impact factor related to the input features, and $s_j$ is a class-specific offset caused by misclassification. Using known data and labels, impact $m$ and offset $s_j$ can be estimated from multiple sets of gradients, and then $\lambda_j$ can be calculated. Ma et al. (2023) transform the label recovery problem into solving a system of linear equations. For each class $j$, they regard $\sigma(z_j) - 1$ and $\sigma(z_j)$ as the coefficients of the target label and the other labels, respectively. By constructing these coefficients into a matrix $A$, they can solve the label vector $y$ from the equation $Ay = \nabla b$, where $\nabla b$ is the gradient w.r.t the bias term of the last layer.

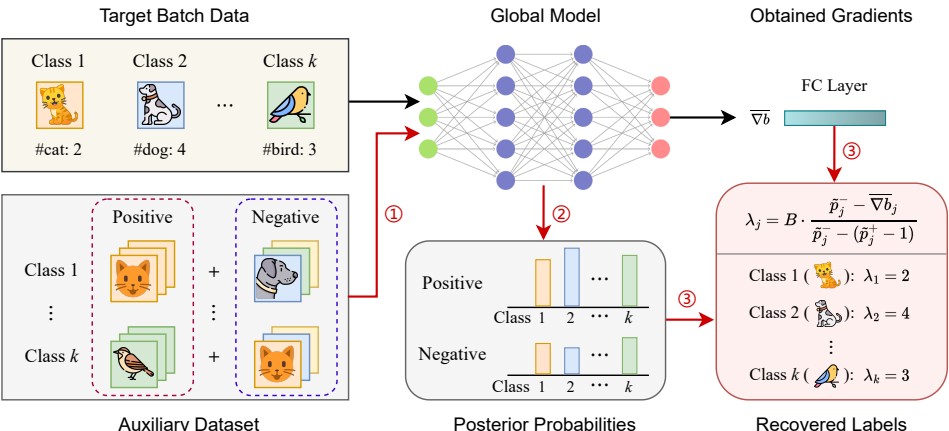

Figure 1: Workflow of our label recovery attack. ① Acquire an auxiliary dataset and divide it into positive and negative subsets for each class. ② Fit the data into the global model for estimating the posterior probabilities of the target batch. ③ Recover the batch labels by substituting the gradients and the posterior probabilities into the derived formula.

## 3 PROBLEM FORMULATION

For a $K$-class classification task in FL, the FedSGD (McMahan et al., 2017) algorithm is used to train the global model $\mathcal{M}$. At a given iteration, a victim client $v$ trains the model with its local batch data $x$ and labels $y$, where $(x, y) \sim \mathcal{D}_v$. Here, $\mathcal{D}_v$ denotes the data distribution of client $v$. Then, the client calculates the batch-averaged gradient $\mathcal{G}$ of the model parameters and sends it to the server. As an *honest-but-curious* server or attacker, we aim to recover the batch labels $y$ from the shared gradient $\mathcal{G}$. The knowledge we have includes the global model $\mathcal{M}$, the shared gradient $\mathcal{G}$, and an auxiliary dataset $x_a$ with the same data distribution as $\mathcal{D}_v$.

In our label attack, we mainly utilize the gradient w.r.t. the bias $b$ in the last fully connected layer, i.e., $\overline{\nabla b}$, to recover the target batch labels $y$. From the auxiliary dataset $x_a$, we randomly sample a portion of instances $\hat{x}_a$, ensuring that the number of instances in each class equals $\pi$. Then we divide $\hat{x}_a$ into positive samples $\hat{x}_j^+$ and negative samples $\hat{x}_j^-$ for each class $j$. By fitting the model $\mathcal{M}$ with these samples, we can obtain the averaged posterior probabilities $\hat{p}^+$ and $\hat{p}^-$ of the positive and negative samples. By substituting $\overline{\nabla b}$, $\hat{p}^+$ and $\hat{p}^-$ into our derived formula in Section 5.2, we can recover the batch labels $\hat{y}$. The workflow of our label attack is shown in Figure 1.

## 4 ESSENCE OF LABEL LEAKAGE

In this section, we investigate the root cause of label leakage from gradients in FL. We first derive a generalized relationship between the gradient $\nabla z$ and the labels $y$, which can be applied to various classification tasks. Then we explain the intrinsic reason for these findings from the perspective of the exponential family. We conclude that the combination of cross-entropy loss and Softmax is intended to reduce the amount of computation, but opens a backdoor to potential attacks.

### 4.1 GENERALIZED EXPRESSION OF GRADIENTS

Without loss of generality, we consider a multi-class classification task using *focal loss* as the loss function and *Softmax* as the activation function. Focal loss is an extension of the cross-entropy loss, which is proposed to solve the problem of class imbalance (Lin et al., 2017). The definition of focal loss in multi-class scenarios can be represented as:

$$\mathcal{L}_{\text{FL}}(p_t) = -\sum_{t=1}^{K} \alpha_t (1 - p_t)^\gamma \log p_t, \tag{1}$$

Table 1: Relationships between different loss function and its gradient $\nabla z$.

| Loss function | $\gamma$ | $\alpha$ | Label | Activation[3] | Gradient $\nabla z$ |
|---|---|---|---|---|---|
| Focal loss | - | - | one-hot | Softmax ($\tau$) | $\frac{1}{\tau}\Phi(\alpha_c, p_c, \gamma)(\boldsymbol{p} - \boldsymbol{y})$ |
| Cross-entropy loss | 0 | 1 | - | Softmax ($\tau$) | $\frac{1}{\tau}(\boldsymbol{p} - \boldsymbol{y})$ |
| Binary cross-entropy loss | 0 | 1 | binary | Sigmoid | $p - y$ |

where $p_t$ is the categorical probability of target class $t$, $\alpha_t$ is the weight of the target class, $\gamma$ is the focusing parameter that controls the degree of class imbalance, and $K$ is the number of classes.

For the $t$th class, $p_t = p_i$ if $i = t$, and $p_t = 1 - p_i$ if $i \neq t$. Here, $p_i$ is the Softmax probability of the $i$th class[1], and $y_i$ is the corresponding label. If we set $\gamma = 0$ and $\alpha_t = 1$, the focal loss is equivalent to the cross-entropy loss $\mathcal{L}_{\text{CE}}(p_i) = -\sum_{t=i}^{K} y_i \log p_i$. Under the same setting, if $K = 2$, $y \in \{0, 1\}$ and the activation is Sigmoid[2], the focal loss is also equivalent to the binary cross-entropy loss $\mathcal{L}_{\text{BCE}}(p) = -y \log p - (1 - y) \log (1 - p)$. Thus, analyzing from focal loss allows us to derive a more general conclusion that can be applied to the other classification variants.

**Theorem 1** (Gradient of Focal Loss). *For a $K$-class classification task using the **focal loss** function and **Softmax** activation, we can derive that the gradient of logit $z_j$ as follows:*

$$\nabla_{z_j}\mathcal{L}_{FL} = \sum_{t=1}^{K} \Phi(\alpha_t, p_t, \gamma) \cdot (p_j - \delta_{tj}), \tag{2}$$

*where $\Phi(\alpha_t, p_t, \gamma) = \alpha_t(1 - p_t)^{\gamma}\left(1 - \gamma\frac{p_t \log p_t}{1 - p_t}\right)$ and $\forall t \in K$, we have $\Phi(\alpha_t, p_t, \gamma) \geq 0$. Besides, $\delta_{tj}$ is the Kronecker delta, which equals 1 if $t = j$ and 0 otherwise.*

*Proof.* See Appendix C.3. □

From Theorem 1, we find that $\nabla_{z_j}\mathcal{L}_{\text{FL}}$ is a summation of $K$ terms, where each term is a product of $\Phi(\alpha_t, p_t, \gamma)$ and $(p_j - \delta_{tj})$. The item $\Phi(\alpha_t, p_t, \gamma)$ can be regarded as the weight of the $t$th class, and $(p_j - \delta_{tj})$ indicates the distance between the Softmax probability of the $j$th class and the target categorical expectation at the $t$th class. In particular, an interesting observation can be made from the latter item is that $(p_j - \delta_{tj})$ is only negative when $t = j$, and positive otherwise, which supports the following conclusions of different loss functions.

From this generalized relationship, we can also derive the gradient of logits $z$ (i.e., $\nabla z$) in other classification variants by setting different values for $\alpha_t$ or $\gamma$, and choosing different label embeddings or activation functions. We summarize the commonly used loss functions, argument settings and the corresponding gradients in Table 1. We can find that the gradient $\nabla z$ is only related to the posterior probabilities $p$ and the target labels $y$. This finding reveals the connection between the gradient $\nabla z$ and the target label $y$, which is the key to label recovery attacks.

## 4.2 EXPLANATION FROM EXPONENTIAL FAMILY

As seen from Table 1, item $(\boldsymbol{p} - \boldsymbol{y})$ exists in different combinations of target labels and activation functions. In particular, if $\Phi(\alpha_c, p_c, \gamma)$ in focal loss is treated as a constant, then gradient $\nabla z$ of each loss function is dominated by $(\boldsymbol{p} - \boldsymbol{y})$. In order to unveil the essential reason for this phenomenon, we explain it from the perspective of the *exponential family* (Andersen, 1970). The exponential family is a class of probability distributions that has the following representation:

$$f_{\text{x}}(\boldsymbol{x}|\boldsymbol{\theta}) = \exp\left[\boldsymbol{\eta}(\boldsymbol{\theta}) \cdot \boldsymbol{T}(\boldsymbol{x}) - A(\boldsymbol{\theta}) + B(\boldsymbol{x})\right], \tag{3}$$

where $\boldsymbol{\theta}$ is the parameter, $\boldsymbol{\eta}(\boldsymbol{\theta})$ is the canonical parameter, $\boldsymbol{T}(\boldsymbol{x})$ is the sufficient statistic, $A(\boldsymbol{\theta})$ is the log-partition function, and $B(\boldsymbol{x})$ is the base measure.

---

[1] $p_i$ is an instance-wise probability, while $p_t$ is a class-wise probability incorporating the target label.

[2] A special case of Softmax with one neuron: $\frac{e^{z_1}}{e^{z_1} + e^{z_2}} = \frac{1}{1 + e^{-(z_1 - z_2)}} = \frac{1}{1 + e^{-z}}$, where $z = z_1 - z_2$.

[3] $\tau$ denotes the temperature parameter in Softmax for softness control.

**Multi-class in Exponential Form.** Take the multi-class classification task as an example, we can build the probability $p(\boldsymbol{x}|\boldsymbol{\theta})$ from the categorical distribution as follows:

$$p(\boldsymbol{x}|\boldsymbol{\theta}) = \prod_{k=1}^{K} \theta_k^{x_k} = \exp\left[\sum_{k=1}^{K} x_k \log \theta_k\right], \tag{4}$$

where $K$ is the number of categories, $x_k \in \{0, 1\}$ and $x_k = 1$ if $x$ belongs to the $k$th category, and $\theta_k$ denotes the probability when $x_k = 1$. Since $\sum_{k=1}^{K} x_k = 1$, we can also derive that $\sum_{k=1}^{K} \theta_k = 1$. Replacing $x_K$ with the first $(K-1)$ items of $x$, we can further express the probability $p(\boldsymbol{x}|\boldsymbol{\theta})$ in the form of the exponential family:

$$\begin{aligned}
p(\boldsymbol{x}|\boldsymbol{\theta}) &= \exp\left[\sum_{k=1}^{K-1} x_k \log \theta_k + \left(1 - \sum_{k=1}^{K-1} x_k\right) \log \theta_K\right] \\
&= \exp\left[\sum_{k=1}^{K-1} x_k \log \frac{\theta_k}{\theta_K} + \log \theta_K\right] \\
&= \exp\left[\boldsymbol{\eta}(\boldsymbol{\theta}) \cdot \boldsymbol{T}(\boldsymbol{x}) - A(\boldsymbol{\theta})\right],
\end{aligned} \tag{5}$$

where $\boldsymbol{\eta}(\boldsymbol{\theta}) = [\eta_1(\boldsymbol{\theta}), \cdots, \eta_{K-1}(\boldsymbol{\theta})]$, $\boldsymbol{T}(\boldsymbol{x}) = [x_1, \cdots, x_{K-1}]$, $A(\boldsymbol{\theta}) = -\log \theta_K$ and $B(\boldsymbol{x}) = 0$.

**Derivation of Softmax and Cross-entropy.** From Equation (5), we obtain $\eta_k = \eta(\theta_k) = \log \frac{\theta_k}{\theta_K}$, which can be transformed into: $\theta_k = e^{\eta_k} \cdot \theta_K$. Incorporating the characteristic of $\sum_{k=1}^{K} \theta_k = 1$, we summarize the $K$ items on both sides of the equation and deduce: $\theta_K \cdot \sum_{k=1}^{K} e^{\eta_k} = \sum_{k=1}^{K} \theta_k = 1$. Hence, $\theta_K$ can be represented as $\theta_K = \frac{1}{\sum_{k=1}^{K} e^{\eta_k}}$. Substituting $\theta_K$ into the relationship between $\theta_k$ and $\eta_k$, we finally derive the Softmax function as follows:

$$\theta_k = e^{\eta_k} \cdot \theta_K = \frac{e^{\eta_k}}{\sum_{j=1}^{K} e^{\eta_j}}. \tag{6}$$

Suppose we focus on an individual sample, so the log-likelihood of sample $x$ is:

$$\ell(\boldsymbol{\theta}; \boldsymbol{x}) = \log p(\boldsymbol{x}|\boldsymbol{\theta}) = \sum_{k=1}^{K} x_k \log \theta_k = -\mathcal{L}_{\text{CE}}(\boldsymbol{x}, \boldsymbol{\theta}), \tag{7}$$

where $\mathcal{L}_{\text{CE}}(\boldsymbol{x}, \boldsymbol{\theta})$ is the cross-entropy loss, which is equivalent to the negative log-likelihood.

Therefore, we find that Softmax has a very strong connection with cross-entropy loss, which can be derived from the categorical distribution in the exponential family. This property interprets why the combination of Softmax and cross-entropy loss is widely used in multi-class classification tasks.

**Gradient of Canonical Parameter.** Given that $A(\boldsymbol{\theta}) = -\log \theta_K$ and $\theta_K = \frac{1}{\sum_{j=1}^{K} e^{\eta_j}}$, we then calculate the derivative of $A(\boldsymbol{\theta})$ w.r.t. $\eta_k$, i.e., $\nabla_{\eta_k} A(\boldsymbol{\theta}) = \frac{e^{\eta_k}}{\sum_{j=1}^{K} e^{\eta_j}} = \theta_k$. We proceed to derive the gradient of the log-likelihood $\ell(\boldsymbol{\theta}; \boldsymbol{x})$ w.r.t. the canonical parameter $\eta_k$ as:

$$\nabla_{\boldsymbol{\eta}} \ell(\boldsymbol{\theta}; \boldsymbol{x}) = \boldsymbol{T}(\boldsymbol{x}) - \nabla_{\boldsymbol{\eta}} A(\boldsymbol{\theta}) = \boldsymbol{T}(\boldsymbol{x}) - \boldsymbol{\theta}. \tag{8}$$

This formula not only exhibits an important feature of the exponential family, but also discloses why the combination of Softmax and cross-entropy loss has the relationship in Table 1. For the exponential family, when parameter $\boldsymbol{\theta}$ is determined, the change of $\ell(\boldsymbol{\theta}; \boldsymbol{x})$ w.r.t. $\boldsymbol{\eta}$ is determined only by $\boldsymbol{T}(\boldsymbol{x})$ and independent of other information about the samples. For a multi-class classification task, the sufficient statistic $\boldsymbol{T}(\boldsymbol{x})$ is actually the target label $\boldsymbol{y}$, the parameter $\boldsymbol{\theta}$ denotes the posterior probabilities $\boldsymbol{p}$, and the canonical parameter $\boldsymbol{\eta}$ implies the logit $\boldsymbol{z}$. Therefore, we can naturally obtain that $\nabla \boldsymbol{z} = \boldsymbol{p} - \boldsymbol{y}$, which is consistent with the previous conclusion.

In summary, the combination of cross-entropy loss and Softmax is derived from the exponential family, which can reduce the amount of computation. However, it also opens a backdoor for potential threats, such as label recovery from gradients in FL.

## 5 LABEL RECOVERY ATTACK

In this section, we propose our label recovery attack. We first present an important observation that the positive (negative) samples of a class have approximate probability distributions. Based on this insight, we exploit an auxiliary dataset for estimating the posterior probabilities of the target batch. Finally, we propose an analytical method that can directly recover the number of labels of each class by substituting the probabilities and gradients into the derived formula.

### 5.1 OUR KEY OBSERVATION

In a multi-class classification task using a neural network, the model first outputs the logits $z$ according to forward propagation, and then normalizes the logits into probabilities $p$ through the Softmax function. By analyzing these posterior probabilities, we empirically observe that different positive (negative) samples of a class have approximate probability distributions.

We carry out the experiments on MNIST and CIFAR10 datasets, which are trained on the LeNet-5 (LeCun et al., 1998) and ResNet-18 (He et al., 2016) models, respectively. To eliminate the influence of the activation function, we choose Tanh for LeNet-5 and ReLU for ResNet-18. For each class, we treat the data belonging to this category as positive samples, and the others as negative samples. Then we aggregate the positive and negative samples from each class during the training procedure to show the correlations and variations in their corresponding probabilities. For ease of demonstration, we only display the results of the first 500 training iterations.

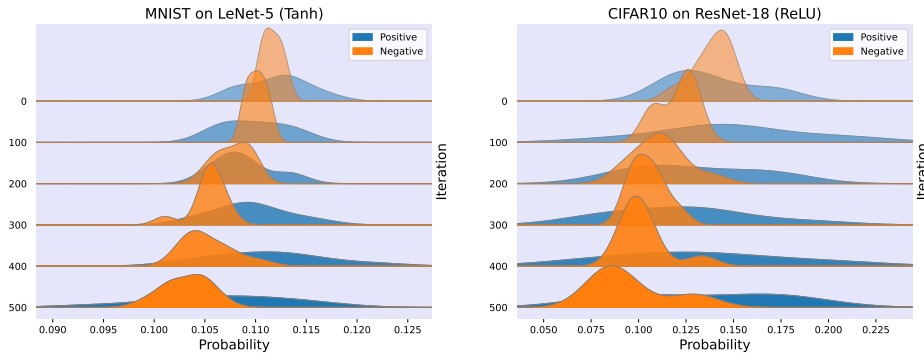

Figure 2: Posterior probability distribution of different iterations.

Figure 2 shows the ridgeline plots of probability distributions from a randomly selected class (class 4 in MNIST and class 7 in CIFAR10) in different iterations. It can be seen that in the initial stages, the positive and negative probabilities are extremely close and have almost the same mean value. As the training progresses, the negative probabilities gradually decrease, while the positive probabilities slowly increase. Although the variance of the probabilities starts to increase, the mean values of the positive or negative probabilities remain within a small range.

We also exhibit the box plots of the probability distributions of all the classes in Figure 3, whose training iteration is 100 for MNIST and 200 for CIFAR10. It is shown that the positive or negative probabilities of each class are gathered around a certain value, although the values are slightly different. For some easily distinguishable classes, the positive and negative probabilities are already separated, such as class 1 in MNIST and class 4 in CIFAR10.

We can interpret the above observations from the model's representation capability. For an untrained model, it cannot discriminate which class the instance belongs to, other than random guesses. Thus, the posterior probabilities of the positive and negative samples are almost the same, equal to $\frac{1}{K}$. As training proceeds, the model gradually learns the data distribution and can distinguish the positive (negative) samples per class. Although the degree of learning varies moderately for different classes, a robust model can give similar confidence scores (i.e., posterior probabilities) for the same class. This explains why the positive (negative) samples per class have similar probability distributions.

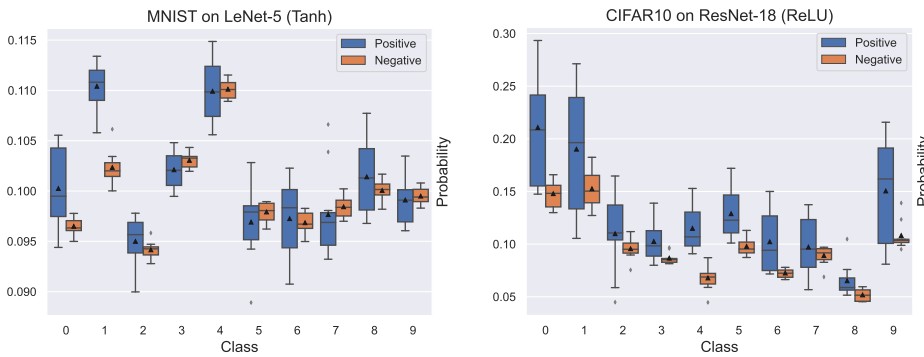

Figure 3: Posterior probability distribution of different classes.

## 5.2 Analytical Label Recovery

Based on the observation in Section 5.1, we can estimate the posterior probabilities of the target batch from an auxiliary dataset, whose data distribution is the same as the training data. Hence, we denote the estimated positive and negative probabilities of the $j$th class as $\hat{p}_j^+$ and $\hat{p}_j^-$, respectively. Combined with the conclusions in Section 4.1, we can derive the following theorem for restoring the batch labels $\lambda_j$ for each class $j$.

**Theorem 2** (Label Recovery Formula). *Having **an auxiliary dataset** with the same distribution of training data, we can recover the class-wise labels $\lambda_j$ in the target batch according to the averaged gradient $\overline{\nabla b}_j$ and the estimated posterior probabilities $\hat{p}_j^+$ and $\hat{p}_j^-$ as follows:*

$$\lambda_j = B \cdot \frac{(\hat{p}_j^- - y_j^-) - \overline{\nabla b}_j/\hat{\varphi}_j}{(\hat{p}_j^- - y_j^-) - (\hat{p}_j^+ - y_j^+)}, \tag{9}$$

*where $y_j^+$ and $y_j^-$ are the pre-set label embeddings of class $j$, $\hat{\varphi}_j = \frac{1}{\tau}\Phi(\alpha_j, \hat{p}_j^+, \gamma)$ is an coefficient related to the $j$th class, and $B$ is the batch size.*

*Proof.* See Appendix C.4. □

Specifically, the label embeddings are pre-defined by the FL protocol, which could be one-hot labels or smoothed labels. For one-hot labels, we have $y_j^+ = 1$ and $y_j^- = 0$. For smoothed labels, we have $y_j^+ = 1 - \epsilon$ and $y_j^- = \frac{\epsilon}{K-1}$, where $\epsilon$ is the smoothing factor. Since $\Phi(\alpha_j, p_j, \gamma)$ is determined by $p_j$, we can use $\hat{p}_j^+$ for replacement and obtain $\hat{\varphi}_j$. By substituting the gradient $\overline{\nabla b}_j$, estimated coefficient $\hat{\varphi}_j$, positive probabilities $\hat{p}_j^+$, $\hat{p}_j^-$ and label embeddings $y_j^+$, $y_j^-$ into the above formula, we can directly recover the number of labels $\lambda_j$ for each class $j$.

## 6 Experiments

We now evaluate our label recovery attack with the prior works on a variety of settings.

### 6.1 Experimental Settings

**Dataset, Model and Activation.** We evaluate our attack on three datasets, including MNIST (LeCun et al., 1998), CIFAR10 (Krizhevsky & Hinton, 2009) and CIFAR100 (Krizhevsky & Hinton, 2009). These datasets are widely used in FL research and cover a variety of classification tasks, such as handwritten digit recognition, object recognition and image classification. We choose LeNet-5 (LeCun et al., 1998), VGG-16 (Simonyan & Zisserman, 2014) and ResNet-50 (He et al., 2016) as the models for the above datasets, respectively. In addition, we select a bunch of activation functions, including Sigmoid, Tanh, ReLU (Nair & Hinton, 2010), ELU (Clevert et al., 2016) and SELU (Klambauer et al., 2017), to verify the universality of our attack.

Table 2: Comparison of our attack with the baselines on diverse scenarios.

| Dataset | Model | Activation | LLG | | iLRG | | Ours | |
|---------|-------|------------|--------|--------|--------|--------|--------|--------|
| | | | ClsAcc | InsAcc | ClsAcc | InsAcc | ClsAcc | InsAcc |
| MNIST | LeNet-5 | Sigmoid | 0.954 | 0.973 | 0.946 | 0.880 | **1.000** | **1.000** |
| | | Tanh | 0.506 | 0.163 | **1.000** | **1.000** | **1.000** | **1.000** |
| CIFAR10 | VGG-16 | ReLU | 0.995 | 0.997 | **1.000** | **1.000** | **1.000** | **1.000** |
| | | ELU | 0.965 | 0.979 | **1.000** | **1.000** | **1.000** | **1.000** |
| CIFAR100 | ResNet-50 | ReLU | 0.937 | 0.952 | **1.000** | **1.000** | **1.000** | **1.000** |
| | | SELU | 0.028 | 0.005 | 0.922 | 0.832 | **0.968** | **0.951** |

**Evaluation Metrics.** To quantitatively evaluate the performance of our label recovery attack, we use the following two metrics: (1) *Class-level label Accuracy (ClsAcc)*: the accuracy measures the proportion of correctly recovered classes; (2) *Instance-level label Accuracy (InsAcc)*: the accuracy measures the proportion of correctly recovered labels in the target batch. In particular, both of these two metrics are realized through Jaccard similarity, which is defined as follows:

$$J(\hat{\mathbf{y}}, \mathbf{y}) = \frac{|\hat{\mathbf{y}} \cap \mathbf{y}|}{|\hat{\mathbf{y}} \cup \mathbf{y}|} = \frac{|\hat{\mathbf{y}} \cap \mathbf{y}|}{|\hat{\mathbf{y}}| + |\mathbf{y}| - |\hat{\mathbf{y}} \cap \mathbf{y}|},$$

where $\hat{\mathbf{y}}$ and $\mathbf{y}$ denote the sets of recovered labels (or classes) and ground-truth labels (or classes).

**Baselines.** Since iDLG (Zhao et al., 2020) only applies to single batch training and non-negative activation functions, we exclude it from the comparison. We mainly compare our attack with LLG (Wainakh et al., 2022) and iLRG (Ma et al., 2023), which do not limit the batch size or activation function. For LLG, we generate the dummy data with the same number as the target batch size and average the results of 10 runs. Since LLG and iLRG are all designed for the untrained models, we mainly compare our attack with them in the untrained setting to be fair.

## 6.2 COMPARISON WITH BASELINES

To exhibit the versatility of our attack, we compare it with the baselines in 3 different groups of settings. We set the batch size to 32 for MNIST and CIFAR10, and 256 for CIFAR100. Without loss of generality, we assume that the training data of each class is uniformly distributed, and the auxiliary dataset is randomly sampled from the validation dataset with 100 samples per class. Furthermore, since the baselines are designed for untrained models, we also use initialized models for comparison to be fair. We run each experiment 20 times and report the average results in Table 2.

From the results, we can see that our attack performs better than the baselines and even achieves 100% ClsAcc and 100% InsAcc in most of the scenarios. In addition, the evaluation results also illustrate that compared with the dataset and model structure, the activation function has a greater impact on the performance of all label recovery attacks. This could be explained by the fact that some

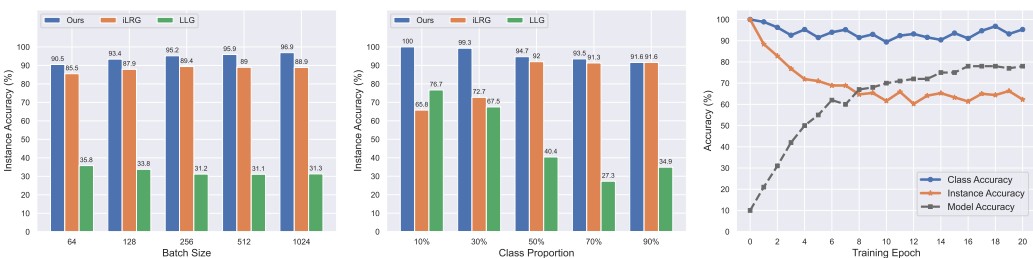

Figure 4: Instance accuracy with different batch sizes and class imbalances.

activation functions, like SELU, produce high variance, which causes the probability distribution of positive and negative samples from the same class to diverge significantly. Therefore, the attack performance of SELU is worse than that of ReLU and ELU. Nevertheless, our attack still shows good results, which demonstrates its effectiveness and universality.

## 6.3 COMPARISON OF VARIOUS FL SETTINGS

From Table 2, it is shown that the attack baselines have the best performance for CIFAR10 on VGG-16 with ReLU activation. So we chose this scenario to analyze the effects of batch size and class imbalance. The batch size varies from 64 to 1024, which is closer to a realistic FL scenario. Class imbalance is a prevalent issue in FL, typically brought on by the unequal distribution of data across various clients. We compare the class proportions from 10% to 90% to simulate the class imbalance. Before launching the attacks, we train the model for 1 epoch with a learning rate of 0.001.

It is shown in Figure 4 that our label recovery is robust to various batch sizes and class imbalance ratios, which maintains over 90% accuracy in all of these settings. As the batch size increases, the InsAcc of our attack gradually improves, which is mainly because the larger the batch size, the more robust the estimation of the averaged posterior probabilities. In addition, since we have the prior distribution of the training data, we can constrain and regularize the estimated labels to improve the success rate of label recovery.

Figure 5: InsAcc with different scales.

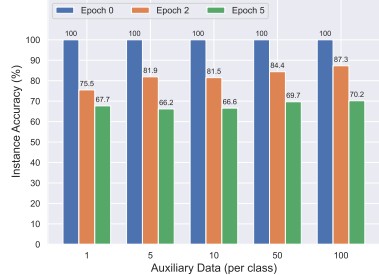

Table 3: InsAcc with classification variants.

| Loss function | Temperature $\tau$ | | Label smoothing $\varepsilon$ | |
|---|---|---|---|---|
| | 0.8 | 1.2 | 0.1 | 0.25 |
| Focal loss | 1.000 | 1.000 | 1.000 | 1.000 |
| Cross-entropy | 1.000 | 1.000 | 1.000 | 1.000 |

## 6.4 ABLATION STUDIES

We conduct ablation studies to analyze the effectiveness of our attack under different classification variants and scales of auxiliary data. Table 3 shows the InsAcc of our attack on an untrained model with the focal loss and cross-entropy loss under different hyper-parameters. The results indicate that our attack is robust to these variants, which can achieve 100% InsAcc in all of these settings. Moreover, we also show the attack performance with different scales of auxiliary data in Figure 5. For an untrained model, the attack performance is not sensitive to the scale of auxiliary data per class, which can achieve 100% InsAcc in all of these settings. For the training models, our attack is slightly affected by the scale but still maintains a reasonable accuracy. This manifests that estimating the posterior probabilities from external data is a robust solution.

## 7 CONCLUSION

In this paper, we explore the potential threats of FL from the perspective of label recovery. We theoretically derive the relationships between gradients and labels in different classification tasks, and then point out the root cause of label recovery from gradients in FL. Based on our key observation that the positive (negative) samples of a class have approximate probability distributions, we propose an analytical method to recover batch labels from the estimated posterior probabilities. Extensive experiments on various datasets and models demonstrate the effectiveness and universality of our attack. For future work, we will design a defense mechanism to mitigate the label leakage according to our theoretical analysis.

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

## A    ETHICS STATEMENT

This work demonstrates vulnerabilities in Federated Learning that could potentially allow malicious actors to exploit gradients to recover private batch labels. If deployed irresponsibly, such analytical attacks could seriously infringe on individuals' privacy and undermine trust in Federated Learning. However, we believe that with proper safeguards and oversight, the insights from this work can be used to improve accountability and integrity.

We recommend developers adopt differential privacy, trusted execution environments, and multi-party computation techniques to help cryptographically secure sensitive information like gradients and batch labels. Rigorous auditing and red team testing should be conducted before deployment to identify and patch vulnerabilities proactively. Policies and procedures governing the appropriate use of model insights should be established, clearly documenting purposes and ensuring transparency.

Furthermore, while we have developed proof-of-concept attacks in a simulated environment, we caution against reckless real-world testing which could cause serious harms. This work is meant to spur improved security practices, not enable adversaries. We advocate for an ethical approach centered on user empowerment and safeguarding rights. If deployed conscientiously with account-ability checks, federated learning can offer privacy-preserving capabilities, but we must be vigilant against misuse. With care, insight and wisdom, we can work towards equitable and trustworthy AI.

## B    RELATED LABEL RECOVERY ATTACKS

We introduce the related label recovery attacks in this section, including iDLG (Zhao et al., 2020), GI (Yin et al., 2021), RLG (Dang et al., 2021), LLG (Wainakh et al., 2022), ZLG (Geng et al., 2021), and iLRG (Ma et al., 2023), and compare them with our proposed label attack.

**iDLG.**    iDLG (Zhao et al., 2020) mathematically derives the relationship between the gradient of the cross-entropy loss w.r.t. the output logits $\nabla z \in \mathbb{R}^K$ and the ground-truth label $y \in \mathbb{R}^K$, which satisfies $\nabla z = p - y$. This relationship reveals that:

- $\nabla z_j$ is negative ($\nabla z_j \in [-1, 0]$) for the input samples belonging to class $j$,
- $\nabla z_j$ is positive ($\nabla z_j \in [0, 1]$) for the samples belonging to other classes $k \neq j$.

Since $\nabla z$ is unavailable in FL, iDLG uses the gradient of the cross-entropy loss w.r.t. the weight $\nabla W$ in the last fully connected layer to estimate $\nabla z$. If the non-negative activation function (e.g., ReLU or Sigmoid) is used in the model, $\nabla W$ has the same sign as $\nabla z$. By summing up the rows of $\nabla W$, the negative row implies the ground-truth class of the target batch. However, iDLG is only applicable to single-batch training and non-negative activation functions.

**GI.**    GI (Yin et al., 2021) follows the main idea of iDLG and extends it to the mini-batch training scenario. GI observes that for an untrained model, the negative values in gradient $\nabla z$ possess larger magnitudes than the positive values, that is $\nabla z_j^- \gg \nabla z_j^+$. This observation indicates that when a training sample of class $j$ is in the target batch, $\sum_{n=1}^{B} \nabla z_j^{(n)}$ is highly probable to be negative, where $B$ is the batch size. Therefore, instead of summing up the rows of $\nabla W$, GI obtains the minimum value in each row of $\nabla W$ and then selects the rows with the minimum $B$ values as the recovered classes of the target batch. However, GI is only applicable to non-repeating classes in the target batch and non-negative activation functions.

**RLG.**    According to iDLG, for a sample belonging to class $j$, $\nabla z_j$ can be distinguished from $\nabla z_{k \neq j}$ by its sign. Since $\nabla z$ can be estimated by $\nabla W$, RLG (Dang et al., 2021) proposes to recover the ground-truth classes of the target batch by distinguishing each row of $\nabla W$ from the other rows. RLG first decomposes $\nabla W^\top$ into $P \Sigma Q$, where $P \in \mathbb{R}^{M \times S}$ and $Q \in \mathbb{R}^{S \times K}$ are orthogonal matrices, and $\Sigma \in \mathbb{R}^{S \times S}$ is a diagonal matrix. For each column $q_j$ of $Q$, $j$ corresponds to the target index $c$ if a hyperplane can be found to separate $q_j$ from the other columns. They transform the problem into finding a classifier to separate $q_{j=c}$ from $q_{j \neq c}$ through linear programming. Although RLG is suitable for all activation functions, it only applies to non-repeating classes in the batch.

**ZLG.** The aforementioned attacks exploit the distinguishability of corresponding rows in $\nabla \boldsymbol{W}$, where the target class is located, to recover the ground-truth classes of the training batch. Since $\nabla \boldsymbol{W} = \nabla \boldsymbol{z} \cdot \boldsymbol{o}^\top = (\boldsymbol{p} - \boldsymbol{y}) \cdot \boldsymbol{o}^\top$, ZLG (Geng et al., 2021) presents to restore batch labels by estimating the posterior probabilities $\boldsymbol{p} \in \mathbb{R}^K$ and input features $\boldsymbol{o} \in \mathbb{R}^M$ of the last layer. ZLG assumes that the summations of different features $\sum_{i=1}^{M} o_i$ are approximately equal to each other, that is $\hat{O} \approx \sum_{i=1}^{M} o_i^{(n)}$ for $n \in [1, B]$. By estimating $\hat{O}$ and $\hat{\boldsymbol{p}}$ from dummy data or auxiliary data, ZLG can restore the number of samples in each class $j$ as follows:

$$\lambda_j = B \left( \hat{p}_j - \frac{1}{\hat{O}} \sum_{i=1}^{M} \nabla \boldsymbol{W}_{j,i} \right).$$

**LLG.** Similar to ZLG, LLG (Wainakh et al., 2022) rewrites $\nabla \boldsymbol{W}$ as $\nabla \boldsymbol{W} = -\boldsymbol{y} \cdot \boldsymbol{o}^\top + \boldsymbol{p} \cdot \boldsymbol{o}^\top$. For each class $j$, the restoration problem is formulated as $\sum_{i=1}^{M} \nabla \boldsymbol{W}_j = \lambda_j m + s_j$, where $\lambda_j$ is the number of labels, $m$ is the impact factor related to the input features, and $s_j$ is a class-specific offset caused by misclassification. Instead of directly estimating the posterior probabilities $\boldsymbol{p}$ and input features $\boldsymbol{o}$, LLG embeds this information into the gradient and indirectly estimates $m$ and $s_j$. By fitting dummy data or auxiliary data into the model and producing multiple sets of gradients, LLG then restores the class-wise labels in the target batch.

**iLRG.** iLRG (Ma et al., 2023) exploits both gradient $\nabla \boldsymbol{W}$ and gradient $\nabla \boldsymbol{b}$ of the bias terms in the last layer to recover the batch labels. According to $\boldsymbol{z} = \boldsymbol{W}\boldsymbol{o} + \boldsymbol{b}$, it is easy to derive that $\nabla \boldsymbol{b} = \nabla \boldsymbol{z}$. Hence, it only needs to estimate the post-softmax probabilities $\boldsymbol{p}$ to recover the batch labels $\boldsymbol{y}$ through the conclusion $\nabla \boldsymbol{b} = \boldsymbol{p} - \boldsymbol{y}$. iLRG first restores the batch averaged features $\bar{\boldsymbol{o}}$ from $\nabla \boldsymbol{W} / \nabla \boldsymbol{b}$ (Geiping et al., 2020) and then calculates the posterior probabilities $\hat{\boldsymbol{p}}$ from $\bar{\boldsymbol{o}}$. For each class $j$, iLRG regards $(p_j - 1)$ and $p_j$ as the coefficients and constructs these coefficients into a matrix $\boldsymbol{A}$. Then it can solve the label occurrence vector $\boldsymbol{\lambda}$ from the equation $\boldsymbol{A}\boldsymbol{\lambda} = \nabla \boldsymbol{b}$.

**Our Attack.** In terms of implementation, our attack leverages the gradient $\nabla \boldsymbol{b}$ and the estimated posterior probabilities to recover the batch labels. At a finer granularity, we divide the probabilities into positive and negative ones for each class $j$, which are denoted as $\boldsymbol{p}_j^+$ and $\boldsymbol{p}_j^-$, respectively. The samples belonging to class $j$ output the $\boldsymbol{p}_j^+$, while the samples belonging to other classes $k \neq j$ output the $\boldsymbol{p}_j^-$. Based on the observation that the positive (negative) samples of a class $j$ have approximate probability distributions, we can estimate the posterior probabilities of the target batch from an auxiliary dataset. The estimated positive and negative probability of the $j$th class are denoted as $\hat{p}_j^+$ and $\hat{p}_j^-$, respectively. Combined with our theoretical deductions, we can directly restore the number of labels $\lambda_j$ for each class $j$ as Equation (9).

## C  DEFINITION AND PROOFS

### C.1  FOCAL LOSS IN MULTI-CLASS CLASSIFICATION

According to the derivation of the binary Focal Loss in (Lin et al., 2017), we extend it into the multi-class scenarios. In a multi-class classification task using Cross-entropy (CE) Loss, the CE loss can be written as follows:

$$\mathcal{L}_{\text{CE}}(\boldsymbol{p}, \boldsymbol{y}) = -\sum_{i=1}^{K} y_i \log(p_i) = \begin{cases} -\log(p_1) & \text{if } y_1 = 1 \\ -\log(p_2) & \text{if } y_2 = 1 \\ \vdots \\ -\log(p_K) & \text{if } y_K = 1, \end{cases}$$

where $\mathbf{y}$ is the one-hot embeded label.

For any class $i$, we use $p_t$ to represent the confidence degree of the model's prediction as the following:

$$p_t = \begin{cases} p_i & \text{if } y_i = 1 \\ 1 - p_i & \text{otherwise,} \end{cases}$$

where $t = i$. To be consistent with the original Focal Loss in (Lin et al., 2017), we use $t$ to represent the class index instead of $i$, and $t$ is actually identical to $i$.

In order to solve class imbalance, Focal Loss assigns an auto-determined weight $(1-p_t)^\gamma$ and a pre-determined weight $\alpha_t$ to each class $t$. Finally, we define the Focal Loss for multi-class classification tasks as:

$$\mathcal{L}_{\text{FL}}(p_t) = -\sum_{t=1}^{K} \alpha_t (1 - p_t)^\gamma \log(p_t).$$

Some important points we would like to emphasize:

- **Difference in probabilities**: $p_i$ is the post-softmax probability, while $p_t$ represents the automatic determined confidence of the input at class $i$, where $t = i$.

- **Mechanism of weight**: For an easy-to-learn sample, $p_t$ might be close to the target label. So, Focal Loss assigns a small coefficient $(1 - p_t)^\gamma$ as the weight. However, for a hard-to-learn sample, $p_t$ may be close to 0. Then $(1 - p_t)^\gamma$ a relatively large weight to enhance the ratio of these samples in the total loss.

- **Summation sign**: Since the binary Focal Loss (Lin et al., 2017) just has one output, the summation is not necessary. In the multi-class case, we use the summation to cover all the classes $t \in [1, K]$, and aim to derive a general conclusion in Theorem 1.

- **Why Focal Loss**: To the best of our knowledge, Focal Loss has the general form in the cross-entropy loss variants and it can be converted to CE loss or BCE loss by setting different $\alpha$ and $\gamma$. We aim to derive a general form of label leakage from gradients, so we choose the Focal Loss.

### C.2 Supplementary Definitions

In a multi-class classification problem, each instance in the dataset belongs to one of several classes. Let's denote the set of classes as $K$ and a particular class of interest as $k \in K$. In this context, we can define positive and negative samples for class $k$.

- **Positive Samples** ($X_k^+$): The positive samples of class $k$ satisfy that: $X_k^+ = \{x_i : y_i = k\}$, where $x_i$ is the input and $y_i$ is the corresponding label.

- **Negative Samples** ($X_k^-$): Similarly, the negative samples of class $k$ satisfy that: $X_k^- = \{x_i : y_i \neq k\}$

According to the positive and negative samples, we can then get the positive and negative probability for class $k$.

- **Positive Probability** ($p_k^+$): When a positive instance is fed into the model, the predicted probability of class $k$ is termed the positive probability. Since the Softmax activation function is used in the output layer, the output posterior probability $p^+$ is a vector of length $k$. Therefore, the positive probability for class $k$ can be expressed as $p_k^+$.

- **Negative Probability** ($p_k^-$): Similarly, when a negative sample is input into the model, the $k$th element of the output probability vector represents the negative probability, denoted as $p_k^-$. It's essential to note that any negative sample associated with the other $(K-1)$ classes contributes to $p_k^-$.

When using an auxiliary dataset to estimate the probabilities of the target training batch in FL, we denote the estimated positive and negative probabilities as $\hat{p}_k^+$ and $\hat{p}_k^-$, respectively.

In a batch size of $B$, we aim to recover the labels of each instance within the batch, i.e., $\mathbf{y} = [y^{(1)}, y^{(2)}, \cdots, y^{(B)}]$. As this is a multi-class classification problem, the ground-truth labels $\mathbf{y}$ can also be represented by the occurrences of each class: $\mathbf{y} = [n_1, n_2, \cdots, n_K]$, where $n_k$ is the number of samples belonging to class $k$ and $K$ is the number of total classes.

- **Class-wise Labels**: The class-wise labels can be defined as: $n_k = \sum_{i=1}^{B} \delta(y^{(i)} = k)$. Here, $n_k$ is the number of samples belonging to class $k$, B is the batch size, $y^{(i)}$ is the true

class label of the $i$th instance in the batch, and $\delta(\cdot)$ is the Kronecker delta function, which equals 1 if the condition inside is true and 0 otherwise.

## C.3 PROOF OF THEOREM 1

**Theorem 3** (Gradient of Focal Loss). *For a $K$-class classification task using the **focal loss** function and **Softmax** activation, we can derive that the gradient of logit $z_j$ as follows:*

$$\nabla_{z_j}\mathcal{L}_{FL} = \sum_{t=1}^{K} \Phi(\alpha_t, p_t, \gamma) \cdot (p_j - \delta_{tj}),$$

*where $\Phi(\alpha_t, p_t, \gamma) = \alpha_t(1-p_t)^\gamma \left(1 - \gamma\frac{p_t \log p_t}{1-p_t}\right)$ and $\forall t \in K$, we have $\Phi(\alpha_t, p_t, \gamma) \geq 0$. Besides, $\delta_{tj}$ is the Kronecker delta, which equals 1 if $t = j$ and 0 otherwise.*

*Proof.* According to Equation (1), we substitute the last $p_t$ with its Softmax formula $p_t = \frac{e^{z_t}}{\sum_{k=1}^{K} e^{z_k}}$, and obtain the transformed focal loss function:

$$\mathcal{L}_{\text{FL}} = -\sum_{t=1}^{K} \alpha_t(1-p_t)^\gamma \log \frac{e^{z_t}}{\sum_{k=1}^{K} e^{z_k}}$$

$$= \sum_{t=1}^{K} \alpha_t(1-p_t)^\gamma \log \sum_{k=1}^{K} e^{z_k} - \sum_{t=1}^{K} \alpha_t(1-p_t)^\gamma z_t.$$

Let $\hbar = (1-p_t)^\gamma$, then we can deduce the gradient of logit $z_j$ as follows:

$$\nabla_{z_j}\mathcal{L}_{\text{FL}} = \sum_{t=1}^{K} \alpha_t \frac{\partial \hbar}{\partial z_j} \log \sum_{k=1}^{K} e^{z_k} + \sum_{t=1}^{K} \alpha_t(1-p_t)^\gamma p_j - \sum_{t=1}^{K} \alpha_t \frac{\partial \hbar}{\partial z_j} z_t - \alpha_j(1-p_j)^\gamma$$

$$= \sum_{t=1}^{K} \alpha_t \frac{\partial \hbar}{\partial z_j} \left(\log \sum_{k=1}^{K} e^{z_k} - z_t\right) + \sum_{t=1}^{K} \alpha_t(1-p_t)^\gamma (p_j - \delta_{tj})$$

$$= \sum_{t=1}^{K} \alpha_t(1-p_t)^\gamma \left(1 - \gamma\frac{p_t \log p_t}{1-p_t}\right)(p_j - \delta_{tj})$$

$$= \sum_{t=1}^{K} \Phi(\alpha_t, p_t, \gamma) \cdot (p_j - \delta_{tj}).$$

$\square$

## C.4 PROOF OF THEOREM 2

**Theorem 4** (Label Recovery Attack). *For the attacker with an **auxiliary dataset**, he can recover the class-wise labels $\lambda_j$ of the target batch according to the averaged gradient $\overline{\nabla}b_j$ and the estimated posterior probabilities $\hat{p}_j^+$ and $\hat{p}_j^-$ as follows:*

$$\lambda_j = B \cdot \frac{(\hat{p}_j^- - y_j^-) - \overline{\nabla}b_j/\hat{\varphi}_j}{(\hat{p}_j^- - y_j^-) - (\hat{p}_j^+ - y_j^+)},$$

*where $y_j^+$ and $y_j^-$ are the pre-set label embeddings of class $j$, $\hat{\varphi}_j = \frac{1}{\tau}\Phi(\alpha_j, \hat{p}_j^+, \gamma)$ is an coefficient related to the $j$th class, and $B$ is the batch size.*

*Proof.* Since $z = Wx + b$, we can deduce that $\nabla b = \nabla z$ and $\overline{\nabla}b_j = \overline{\nabla}z_j$. We expand the averaged gradient $\overline{\nabla}z_j$ as a summation of $B$ terms and replace the posterior probability $p_j^{(n)}$ of each sample

$n$ with its estimated probabilities $\hat{p}_j^+$ and $\hat{p}_j^-$. Because $\Phi(\alpha_j, p_j^{(n)}, \gamma)$ is only related to the positive samples of the $j$th class, we can replace $p_j^{(n)}$ with $\hat{p}_j^+$. So we have:

$$\hat{\varphi}_j = \frac{1}{\tau}\Phi(\alpha_j, \hat{p}_j^+, \gamma) = \frac{1}{\tau}\alpha_j(1 - \hat{p}_j^+)^\gamma \left(1 - \gamma\frac{\hat{p}_j^+ \log \hat{p}_j^+}{1 - \hat{p}_j^+}\right).$$

Assume that the first $\lambda_j$ samples belong to the $j$th class, and the rest $(B - \lambda_j)$ samples belong to other classes. Then from the first row of Table 1, we can derive that:

$$\overline{\nabla b}_j = \frac{1}{B}\sum_{n=1}^{B}\nabla b_j^{(n)} = \frac{1}{B}\left\{\sum_{n=1}^{\lambda_j}\varphi_j^{(n)}\left[p_j^{(n)} - y_j^{(n)}\right] + \sum_{n=\lambda_j+1}^{B}\varphi_j^{(n)}\left[p_j^{(n)} - y_j^{(n)}\right]\right\}$$

$$\approx \frac{1}{B}\left\{\lambda_j\,\hat{\varphi}_j\left(\hat{p}_j^+ - y_j^+\right) + (B - \lambda_j)\hat{\varphi}_j\left(\hat{p}_j^- - y_j^-\right)\right\}.$$

Therefore, we can finally derive that:

$$\lambda_j = B \cdot \frac{\hat{\varphi}_j\left(\hat{p}_j^- - y_j^-\right) - \overline{\nabla b}_j}{\hat{\varphi}_j\left(\hat{p}_j^- - y_j^-\right) - \hat{\varphi}_j\left(\hat{p}_j^+ - y_j^+\right)} = B \cdot \frac{(\hat{p}_j^- - y_j^-) - \overline{\nabla b}_j/\hat{\varphi}_j}{(\hat{p}_j^- - y_j^-) - (\hat{p}_j^+ - y_j^+)}.$$

$\square$

## C.5 INTERPRETATION OF THEORETICAL ANALYSIS FROM EXPONENTIAL FAMILY

The standard form of exponential family distribution is:

$$f_{\mathrm{x}}(\boldsymbol{x}|\boldsymbol{\theta}) = \exp\left[\boldsymbol{\eta}(\boldsymbol{\theta}) \cdot \boldsymbol{T}(\boldsymbol{x}) - A(\boldsymbol{\theta}) + B(\boldsymbol{x})\right].$$

We know that the likelihood is the joint probability of all samples occurring:

$$\begin{aligned}
L(\boldsymbol{\theta}; \boldsymbol{x}) &= f(\boldsymbol{x}_1, \ldots, \boldsymbol{x}_N|\boldsymbol{\theta}) \\
&= \prod_{i=1}^{N} f(\boldsymbol{x}_i|\boldsymbol{\theta}) \\
&= \prod_{i=1}^{N} \exp\left[\boldsymbol{\eta}(\boldsymbol{\theta}) \cdot \boldsymbol{T}(\boldsymbol{x}_i) - A(\boldsymbol{\theta}) + B(\boldsymbol{x}_i)\right] \\
&= \exp\left[\boldsymbol{\eta}(\boldsymbol{\theta}) \cdot \sum_{i=1}^{N}\boldsymbol{T}(\boldsymbol{x}_i) - NA(\boldsymbol{\theta}) + \sum_{i=1}^{N}B(\boldsymbol{x}_i)\right].
\end{aligned}$$

Now let's add logarithms to the likelihood function to get the log-likelihood function:

$$\begin{aligned}
\ell(\boldsymbol{\theta}; \boldsymbol{x}) &= \log L(\boldsymbol{\theta}; \boldsymbol{x}) \\
&= \boldsymbol{\eta}(\boldsymbol{\theta}) \cdot \sum_{i=1}^{N}\boldsymbol{T}(\boldsymbol{x}_i) - NA(\boldsymbol{\theta}) + \sum_{i=1}^{N}B(\boldsymbol{x}_i).
\end{aligned}$$

For the exponential family, parameter $\boldsymbol{\eta}$ and $\boldsymbol{\theta}$ are reversible. Hence, the derivative of canonical parameter $\boldsymbol{\eta}$ is denote as:

$$\nabla_{\boldsymbol{\eta}}\ell(\boldsymbol{\theta}; \boldsymbol{x}) = \sum_{i=1}^{N}\boldsymbol{T}(\boldsymbol{x}_i) - N\nabla_{\boldsymbol{\eta}}A(\boldsymbol{\theta}).$$

For the categorical distribution, we know that $\nabla_{\boldsymbol{\eta}} A(\boldsymbol{\theta}) = \boldsymbol{\theta}$, $\boldsymbol{T}(\boldsymbol{x}) = \boldsymbol{x}$ and $\mathcal{L}_{\text{CE}}(\boldsymbol{\theta}, \boldsymbol{x}) = -\ell(\boldsymbol{\theta}; \boldsymbol{x})$. Then we can derive the final expression in the following:

$$\nabla_{\boldsymbol{\eta}} \mathcal{L}_{\text{CE}}(\boldsymbol{\theta}, \boldsymbol{x}) = -\nabla_{\boldsymbol{\eta}} \ell(\boldsymbol{\theta}; \boldsymbol{x})$$

$$= N \nabla_{\boldsymbol{\eta}} A(\boldsymbol{\theta}) - \sum_{i=1}^{N} \boldsymbol{T}(\boldsymbol{x}_i)$$

$$= N \boldsymbol{\theta} - \sum_{i=1}^{N} \boldsymbol{x}_i.$$

From this formula, we can observe that the characteristic of the Exponential Family gives an efficient way for CE loss to calculate the loss of canonical parameter $\boldsymbol{\eta}$ by just doing a subtraction rather than complex calculations. In the multi-class scenario, after substituting $\boldsymbol{\theta}$ with post-softmax probability $\boldsymbol{p}$, $\boldsymbol{\eta}$ with logits $\boldsymbol{z}$, and $\boldsymbol{x}_i$ with target label $\boldsymbol{y}_i$, we can derive $\nabla_{\boldsymbol{z}} \mathcal{L}_{\text{CE}}$ (reduction in summation) as follows:

$$\nabla_{\boldsymbol{z}} \mathcal{L}_{\text{CE}}(\boldsymbol{p}, \boldsymbol{y}) = N \boldsymbol{p} - \sum_{i=1}^{N} \boldsymbol{y}_i.$$

This is what we mean that from an exponential family perspective, the combination of the softmax and cross-entropy would have reduced computation, but opened a back door to leaking labels from shared gradients.

## D  ABLATION STUDIES

### D.1  ATTACK ON FOCAL LOSS

In this section, we present additional experiments conducted on the Focal Loss. We mainly test the parameters of $\tau$, $\gamma$ and $\varepsilon$ on an untrained model, and average the experiments over 10 trials. Focusing on temperature $\tau$, we present several cases where the accuracy is not 100%. In addition, by varying $\gamma$ and $\varepsilon$ on these settings, the accuracies are not affected. The following table shows the ClsAcc and InsAcc of our attack on the Focal Loss with different temperatures $\tau$ (batch size=64, activation=ReLU).

Table 4: Label recovery accuracies on Focal Loss ($\gamma = 2$, $\epsilon = 0$).

| $\tau$ | MNIST (LeNet) | | CIFAR-10 (ResNet-18) | | CIFAR-100 (ResNet-50) | |
|---|---|---|---|---|---|---|
| | Our ClsAcc | Our InsAcc | Our ClsAcc | Our InsAcc | Our ClsAcc | Our InsAcc |
| **0.5** | 0.980 | 0.906 | 0.990 | 0.960 | 1.000 | 1.000 |
| **0.75** | 0.980 | 0.945 | 1.000 | 0.994 | 1.000 | 1.000 |
| **0.9** | 0.990 | 0.954 | 1.000 | 1.000 | 1.000 | 1.000 |
| **1.25** | 0.990 | 0.983 | 1.000 | 1.000 | 1.000 | 1.000 |
| **1.5** | 1.000 | 0.998 | 1.000 | 1.000 | 1.000 | 1.000 |

We have observed that the temperature parameter, $\tau$, significantly impacts smaller datasets. When $\tau$ is smaller, the space of logits expands, complicating the estimation of batch posterior probabilities. Consequently, as $\tau$ decreases, label accuracy deteriorates. For the large datasets with more classes, such as CIFAR-10, the logit space is hardly influenced by changing different $\tau$.

### D.2  AUXILIARY DATASET WITH DIFFERENT ATTACKS

In this section, we compare the label recovery accuracies of ZLG, LLG and our attack on different settings. We use the training dataset to sample the target batch for label recovery and the validation

dataset to simulate the auxiliary dataset. This ensures that the auxiliary dataset has the same data distribution as the training dataset. We compare the label recovery accuracies (ClsAcc and InsAcc) of different attacks on three groups of experiments, averaging results over 10 trials.

The first group of experiments set the batch size to 64 and the activation function to ReLU. The second group of experiments set the batch size to 256 and the activation function to ReLU. Finally, the third group of experiments set the batch size to 64 and the activation function to ELU. The results are presented in the following tables.

Table 5: Label recovery accuracies (batch size=64, activation=ReLU).

| Dataset | Model | ZLG | | LLG | | Our | |
|---------|-------|--------|--------|--------|--------|--------|--------|
| | | ClsAcc | InsAcc | ClsAcc | InsAcc | ClsAcc | InsAcc |
| MNIST | LeNet | **1.000** | 0.996 | **1.000** | **1.000** | 0.995 | 0.955 |
| CIFAR-10 | LeNet | **1.000** | 0.986 | **1.000** | **0.988** | **1.000** | 0.969 |
| CIFAR-10 | ResNet-18 | **1.000** | 0.916 | **1.000** | 0.914 | **1.000** | **1.000** |
| CIFAR-100 | ResNet-50 | 0.985 | 0.893 | 0.816 | 0.633 | **1.000** | **1.000** |

Table 6: Label recovery accuracies (batch size=256, activation=ReLU).

| Dataset | Model | ZLG | | LLG | | Our | |
|---------|-------|--------|--------|--------|--------|--------|--------|
| | | ClsAcc | InsAcc | ClsAcc | InsAcc | ClsAcc | InsAcc |
| MNIST | LeNet | **1.000** | 0.977 | **1.000** | **0.982** | **1.000** | 0.951 |
| CIFAR-10 | LeNet | **1.000** | 0.961 | **1.000** | **0.970** | **1.000** | 0.956 |
| CIFAR-10 | ResNet-18 | **1.000** | 0.905 | **1.000** | 0.919 | **1.000** | **0.977** |
| CIFAR-100 | ResNet-50 | 0.998 | 0.881 | **1.000** | 0.868 | **1.000** | **1.000** |

Table 7: Label recovery accuracies (batch size=64, activation=ELU).

| Dataset | Model | ZLG | | LLG | | Our | |
|---------|-------|--------|--------|--------|--------|--------|--------|
| | | ClsAcc | InsAcc | ClsAcc | InsAcc | ClsAcc | InsAcc |
| MNIST | LeNet | **1.000** | 0.948 | 0.580 | 0.283 | 0.990 | **0.969** |
| CIFAR-10 | LeNet | **1.000** | 0.923 | 0.990 | 0.743 | **1.000** | **0.939** |
| CIFAR-10 | ResNet-18 | **1.000** | 0.902 | 0.980 | 0.697 | **1.000** | **0.972** |
| CIFAR-100 | ResNet-50 | 0.985 | 0.897 | 0.845 | 0.603 | **1.000** | **1.000** |

From these tables, we can observe that our attack is more robust than ZLG and LLG on complex datasets like CIFAR-10 or CIFAR-100 trained with ResNet-18 or ResNet-50.

- For the LeNet model with ReLU activation, when the batch size increases from 64 to 256, all the attacks have a slight decrease in InsAcc. It is straightforward to understand that a larger batch size is much more difficult to attack than a smaller batch size.

- For the CIFAR-10 dataset trained on LeNet or ResNet-18, we can observe that the InsAcc of ZLG and LLG is lower than our attack. When changing the shallow LeNet model to the deep ResNet-18 model, the InsAcc of ZLG and LLG decreases significantly. However, the InsAcc of our attack is improved, which indicates that our attack is more robust than ZLG and LLG on deep neural networks.

- For the CIFAR-100 dataset, it is shown that our attack reaches 100% InsAcc on all the settings. We conclude that a dataset with more classes makes it easy for our attack to recover the labels. This is because the more classes there are, the estimated probabilities are more accurate and concentrated around the ground-truth value.

- Finally, when changing the activation function from ReLU to ELU, the InsAcc of LLG drops dramatically. This is because LLG is based on the assumption that the activation function is non-negative. When the activation function is changed to ELU, the assumption is violated, and the InsAcc of LLG drops significantly.

### D.3  AUXILIARY DATASET WITH DISTRIBUTION SHIFT

To address concerns about significant distribution shifts, we conducted supplementary experiments, averaging results over 10 trials.

In the first set of experiments, we alternately used CIFAR-10 and CINIC-10 (Darlow et al., 2018) as the training and auxiliary datasets. CINIC-10 extends CIFAR-10 by incorporating downsampled ImageNet images. While there is some overlap in the distributions due to similar object categories, there is a notable bias in the data features. We employed an untrained VGG model with a batch size of 64.

Similarly, the second set of experiments involves alternating between MNIST and Fashion-MNIST as the training and auxiliary datasets. Despite both datasets having 10 classes, the objects they represent are entirely different. MNIST dataset contains a lot of handwritten digits, while Fashion-MNIST represents the article of clothing. Employing an untrained LeNet model with a batch size of 64, the results are presented in the table below.

Table 8: Distribution shift between training dataset and auxiliary dataset.

| Training Dataset | Auxiliary Dataset | Model | Our ClsAcc | Our InsAcc |
|---|---|---|---|---|
| CIFAR-10 | CINIC-10 | VGG | 1.000 | 1.000 |
| CINIC-10 | CIFAR-10 | VGG | 1.000 | 1.000 |
| MNIST | FMNIST | LeNet | 1.000 | 0.969 |
| FMNIST | MNIST | LeNet | 1.000 | 0.948 |

This phenomenon can be explained as follows: In the initial phase of model training, the model lacks the ability to differentiate between samples from each class, assigning similar output probabilities to all fitted samples (i.e., $1/K$). For different training datasets like Fashion-MNIST and MNIST, the model projects input figures to similar output probabilities with slight variations. This benefits our label recovery attack as it becomes easier to estimate the posterior probabilities of the target batch.

Given that Fashion-MNIST is more intricate than MNIST, utilizing MNIST as the auxiliary dataset poses challenges in accurately estimating the probability distribution of batch training samples. This difficulty accounts for the marginal decrease in InsAcc. However, since CIFAR-10 and CINIC-10 exhibit similarities, there is no difference in InsAcc. The outcomes of these experiments illustrate that if an attacker initiates an attack during the early training stages, having an auxiliary dataset with an identical distribution to the training dataset may not be essential.

