# OpenReview forum: "Posterior Probability-Based Label Recovery Attack in Federated Learning"
_ICLR.cc/2024/Conference — Submitted to ICLR 2024_

### Official Review · Reviewer_gCmo · 2023-10-17

**Soundness:** 2 fair
**Presentation:** 2 fair
**Contribution:** 2 fair
**Rating:** 6
**Confidence:** 5

**Summary:**

The authors propose a new method for extracting batch label counts from gradients. The method uses auxiliary data to estimate the network probabilities for data coming from different classes and uses that knowledge to approximate the batch label counts. Additionally, the authors show that their framework is more generic and allows attacking more than the softmax + CE losses used in previous work, instead also handling the focal family of losses. Finally, they provide a theoretical analysis of the label leakage problem through the lens of the exponential family. They demonstrate that their algorithm recovers label counts efficiently in several settings, including under class imbalance, label smoothing, different network activations, different model depths, and different batch sizes.

**Strengths:**

- Good practical results - beats SOTA everywhere
- Experiments  with many different settings on the CE + softmax loss
- Provides extension of the method to the focal loss - not considered before in the literature
- Possibly an interesting idea to look at the exponential family and their respective losses to explain label leakage, however, the analysis is currently too simplistic

**Weaknesses:**

- **The focal loss**:
The focal loss definition is confusing. It doesn't seem to be internally consistent throughout the paper, and it does not seem to be exactly following [1] either. This makes it very hard to check the associated maths. In particular, in Eq. 1, $t$ is just an index of the sum, while in the first paragraph of page 4, it is the "target class." Further, the sum over $t$ in Eq. 1 does not appear in the original description in [1]. As currently defined, where $y$ appears in Eq. 1 is not clear - according to the text on page 4, it is embedded in $p_t$ (in a different way from how it is embedded in [1]), but according to the proof in Appendix A.1 that is not the case and $p_t$ is simply the post-softmax probability. According to [1], $\alpha_t$ depends on $y$, but the paper specifies nothing about this. Further, the temperature parameter $\tau$ and the smoothed labeling options are only passingly defined. Finally, in the proof in Appendix A.2, the sum over $t$ that is given in Eq. 1 seems to disappear.
The problem is further exacerbated by inconsistent notations. In particular, $i$s and $t$s are used interchangeably in the first paragraph on page 4, even though the authors themselves claim to differentiate them. For example, see the definition of $p_t$, and $\mathcal{L}_\text{CE} (p_t)$ .
Finally, in [1], the focal loss is defined in the case of multiple foreground classes and a single background class, which, to my understanding, are treated slightly differently from each other and which also results in loss slightly different than $\mathcal{L} _\text{CE} (p_t)$ when $\gamma=0$ and $\alpha_t=1$. Do the authors consider the same setup or not, and how are these discrepancies resolved?
 - **Theoretical analysis of the label leakage:**
The authors claim their analysis gives valuable insights regarding the origin of label leakage. I find their analysis falls very short from this promise for a few reasons.
On the one hand, the derivation of the gradient of the network loss is not done for the full exponential family but only for a single member (the exponential probability that gives rise to the softmax and cross-entropy loss), jeopardizing the generality of the proposed conclusions.
On the other hand, even in that one case, the proposed derivations are not novel, as they follow directly from plugging the results of the standard theory for deriving the cross entropy loss and the softmax, which the authors should more explicitly refer to, into the gradient of these functions which have been derived in the context of label leakage multiple times ( as the authors acknowledge in their background section ).
This leaves as possible contributions of the authors' analysis only describing the gradient of the logits of softmax in the context of the exponential family notions and definitions. This is also not a strong contribution, however, as the authors do not spend more than two sentences on this interpretation. In particular, the authors do not discuss how computation is saved by the exponential family at all and do not explain how the exponential family poses a privacy threat beyond the CE + softmax combination.
- **Missing Comparisons and Citations:**
1. The authors should cite [3] and [4] as relevant prior work on label leakage. They are both very relevant as [4] works on non-positive activations similarly to this work, while [3] talks about the possibility of using auxiliary data to estimate quantities that can be used to estimate the label counts. I think comparing to [3] will be good also.
2. The paper will benefit from better discussion about differences to prior work and, in particular, [2], [3], and [5]. All these works, similarly to the proposed method, estimate a quantity that they later plug into $p - y$ to compute the label counts $\lambda_j$. Further, at least [3] and [5] talk about using auxiliary data to do so, again similarly to this work.
3. In the background section, the paper can benefit from discussing the how auxiliary data have been used in gradient inversion attacks before - e.g. [8-11]
- **Proposed additional evaluations and missing evaluation details:**
1. What dataset/model/batch size/epoch/label distribution is used in Table 3? Are $\tau$ and $\epsilon$ assumed to be known by the attacker? Can you redo the experiments in a setting where we do not get 100% accuracy?
2. What is the performance if there is a distribution shift between the auxiliary data used for label recovery and the client data?
3. In Sec 6.1/6.2, do you use the focal or cross-entropy loss?
4. [2] has a proposed version that works on models trained for several epochs. The authors should compare their method to it.
5. [3] and [5] can be applied using auxiliary data. The authors should compare those methods to the proposed method under the same auxiliary data.
6. Why in Sec. 6.3 do the authors use models trained for one epoch?
7. In Sec. 6.3 the authors say, "Since we have the prior distribution of the training data, we can constrain and regularize the estimated labels to improve the success rate of label recovery." Is that something the authors do in their experiments, and if so, how exactly?
8. What label distributions do the authors use in 6.1? Uniform at random?
- **Typos and other Nits**:
1. Figure 2 will benefit from a box plot instead of the current figure, as the variance of the positives is huge, and it becomes hard to judge where means/medians are.
2. Bolding in Table 2 should be applied to all 1.000, not only the proposed method for consistency.
3. On the first line of the derivation of $\nabla_{z_j}\mathcal{L}_{FL}$ in Appendix A.1, there is missing $z_t$ in the last sum
4. On the last line of the derivation of $\nabla_{z_j}\mathcal{L}_{FL}$ in Appendix A.1, $\Psi\rightarrow \Phi$
5. Second summand in the second line of the derivation of $\nabla b_j$ in Appendix A.2, $(B-\lambda)\rightarrow(B-\lambda_j )$
6. First part of the derivation of $\lambda_j$  in Appendix A.2, there is no division by $\varphi_j$
- **Other:**
The terminology used in the paper is slightly non-standard and this makes the abstract and intro hard to read. In particular, without reading deep into the paper, it is not clear what is meant by "classification variants" - the authors can just say different classification losses and network activation functions. Further, talking about posterior probability distributions is a bit confusing too - you can just call them the probabilities predicted by the network or post-softmax probabilities. Finally, "approximate probability distributions" do not clearly convey that you are talking about approximating from data the distributions of the elements of the output of the softmax.

**Questions:**

- Can the authors fix consistently throughout the paper and appendix the definition and usage of the focal loss?
- Can the authors provide an extension of the current theoretical analysis to the general exponential family or at least to the subset of the exponential family covering the focal loss?
- Can the authors expand the discussion about the interpretation of their theoretical findings, in particular explaining in more detail their computational and privacy arguments in the context of the full exponential family?
- Can the authors expand the focal loss experiments to more settings, especially ones where the success rate is not 100%? For a paper that focuses on proposing a generic method that works on multiple losses that are subversions of the focal loss, the paper provides surprisingly few experiments with that loss.
- Can the authors provide a discussion about why the focal loss is significant in particular? Are there other common classification losses that it covers as sub-cases? Are there other common classification losses not covered by the focal loss that might be more secure?
- Can the authors provide a comparison with [2] for networks trained for several epochs?
- Can the authors provide a discussion of how their method compares to prior work - e.g. [2,3,5]? All these methods and the author's proposed method in practice approximate some quantities related to $p_t$ and plug them in the same formula to get $\lambda_j$. Thus, as of now, I have no idea why the author's proposed approximation is better than prior work.
- Can the authors provide a comparison with [3] or [5] when auxiliary data is used for estimating their parameters?
- Can the authors test their methods in a FedAvg setting (check [3,6,7])?
- Do the authors know why the SELU activation results in so much more variance in the estimates of its posterior probability distribution?
- Can the authors provide the missing information mentioned in the weakness section?
- Can the authors provide code?

All in all, I find the experimental part of this paper strong. Thus, I am leaning toward acceptance. However, the focal loss definition problem is significant, as it makes it hard to fully check the mathematics. Further, the theoretical analysis in its current form does not contribute to the paper despite its interesting idea. Additionally, I really want to see a discussion about the differences to prior work ( discussion also sorely missed in prior work ), as many works in the field are slight variations of each other, and it is hard to understand where improvements between the papers, including this one, comes from. Finally, additional discussion of why the focal loss matters and more experiments with it just make a lot of sense in the context of this paper as it is one of the biggest claimed contributions.

[1] Tsung-Yi Lin, Priya Goyal, Ross Girshick, Kaiming He, and Piotr Dollar. Focal loss for dense object detection. In Proceedings of the IEEE International Conference on Computer Vision, pp. 2980–2988, 2017.
[2] Kailang Ma, Yu Sun, Jian Cui, Dawei Li, Zhenyu Guan, and Jianwei Liu. Instance-wise batch label restoration via gradients in federated learning. In The Eleventh International Conference on Learning Representations, 2023.
[3] Geng, Jiahui, et al. "Towards general deep leakage in federated learning." arXiv preprint arXiv:2110.09074 (2021).
[4] Trung Dang, Om Thakkar, Swaroop Ramaswamy, Rajiv Mathews, Peter Chin, and Franc¸oise Beaufays. Revealing and protecting labels in distributed training. Advances in Neural Information Processing Systems, 34:1727–1738, 2021.
[5] Aidmar Wainakh, Fabrizio Ventola, Till Mußig, Jens Keim, Carlos Garcia Cordero, Ephraim Zimmer, Tim Grube, Kristian Kersting, and Max Muhlh ¨ auser. User-level label leakage from gradients in federated learning. Proceedings on Privacy Enhancing Technologies, 2:227–244, 2022.
[6] Dimitrov, Dimitar Iliev, et al. "Data leakage in federated averaging." Transactions on Machine Learning Research (2022).
[7] Zhu, Junyi, Ruicong Yao, and Matthew B. Blaschko. "Surrogate model extension (SME): A fast and accurate weight update attack on federated learning." arXiv preprint arXiv:2306.00127 (2023).
[8] Zhuohang Li, Jiaxin Zhang, Luyang Liu, and Jian Liu. Auditing privacy defenses in federated learning via generative gradient leakage. In Proceedings of the IEEE/CVF Conference on Computer Vision and Pattern Recognition, pages 10132–10142, 2022.
[9] Wu, Ruihan, et al. "Learning To Invert: Simple Adaptive Attacks for Gradient Inversion in Federated Learning." Uncertainty in Artificial Intelligence. PMLR, 2023.
[10] Dongyun Xue, Haomiao Yang, Mengyu Ge, Jingwei Li, Guowen Xu, and Hongwei Li. Fast generation-based gradient leakage attacks against highly compressed gradients. IEEE INFO316 COM 2023 - IEEE Conference on Computer Communications, 2023.
[11] Garov, Kostadin, et al. "Hiding in Plain Sight: Disguising Data Stealing Attacks in Federated Learning." arXiv preprint arXiv:2306.03013 (2023).

**Details Of Ethics Concerns:**

The authors should include an ethics statement discussing how their work can be used by malicious actors and how to defend against it.

---

> ### Comment · Reviewer_gCmo · 2023-11-22
> **Response to Authors**
>
> Dear authors,
>
> I realize my review is voluminous and, therefore, takes a while to answer. However, as the end of the review period is coming really soon, I would encourage you to post a partial answer to my review so that we can engage in discussion while finishing the rest of your response.
>
> Thank you,
> Reviewer gCmo

---

> ### Author Response · Authors · 2023-11-22
> **Response to Reviewer gCmo**
>
> We greatly appreciate your meticulous reading this paper and providing so many valuable suggestions. Regardless of the final result, we are very lucky to have a reviewer like you! We carefully read your comments and rethink about how to improve this work. We will try our best to answer your questions and concerns. We are pleased to offer more explanation or revisions if you have any other recommendations.
>
> > Q1: Can the authors fix consistently throughout the paper and appendix the definition and usage of the focal loss?
>
> In Appendix A.1, we step-by-step derive the Focal Loss in multi-class scenarios from the original binary Focal Loss in [1].
>
> Some important points we would like to emphasize:
>
>
> - **Difference in probabilities**: $p_i$ is the post-softmax probability, while $p_t$ represents the automatic determined confidence of the input at class $i$, where $t=i$.
> - **Mechanism of weight**: For an easy-to-learn sample, $p_t$ might be close to the target label. So, Focal Loss assigns a small coefficient $(1-p_t)^{\gamma}$ as the weight. However, for a hard-to-learn sample, $p_t$ may be close to 0. Then $(1-p_t)^{\gamma}$ a relative large weight to enhance the ratio of the these samples in the total loss.
> - **Summation sign**: Since the binary Focal Loss [1] just has one output, the summation is not necessary. In the multi-class case, we use the summation to cover all the classes $t\in[1,K]$, and aim to derive a general conclusion in Theorem 1.
> - **Why Focal Loss**: To the best of our knowledge, Focal Loss has the general form in the cross-entropy loss variants and it can be converted to CE loss or BCE loss by setting different $\alpha$ and $\gamma$. We aim to derive a general form of label leakage from gradient, so we choose the Focal Loss.
>
> [1] Tsung-Yi Lin, Priya Goyal, Ross Girshick, Kaiming He, and Piotr Dollar. Focal loss for dense object detection. In Proceedings of the IEEE International Conference on Computer Vision, pp. 2980–2988, 2017.
>
> > Q2: Can the authors provide an extension of the current theoretical analysis to the general exponential family or at least to the subset of the exponential family covering the focal loss?
>
> We would like to explain the reason of introducing the Exponential Family. Actually, we introduce the Exponential Family to better understand **why the combination of cross-entropy and normalization functions (softmax / sigmoid) leads to the conclusions of Theorem 1** in this paper. Although the studies like iDLG have deduced some conclusions about cross-entropy and softmax, they do not delve into the underlying reasons. For illustration, we also find that sigmoid + cross-entropy also satisfies this conclusion. **We also wonder if there are other functions that can replace softmax and whether the label leakage still holds?** So we ask ourselves the following three questions to:
>
> 1. How did softmax (or sigmoid) originate and why is it applied to classification problems?
> 2. Why can cross-entropy serve as a loss function for classification problems?
> 3. When softmax and cross-entropy are combined, why does the term $(\boldsymbol p-\boldsymbol y)$ appear?
>
> The exponential family can provide satisfactory answers to these questions. The Focal Loss is a variant of the common CE loss, and its unique term $(1-p_t)^{\gamma}$ represents the auto-determined weight for class imbalance. We find that from the perspective of exponential family, the cross-entropy loss and softmax (or sigmoid) are naturally combined with each other. **To this end, we make sure that no other functions can replace softmax (or sigmoid), and the occurence of label leakage is in evitable in a classification model shared gradients.**
>
> The Exponential Family also encompasses the Bernoulli, Poisson and Gaussian distributions. However, these distributions do not support the conclusion proposed in this paper. Therefore, the conclusion is only applicable to the Categorical distribution within the Exponential Family.
>
> > Q3: Can the authors expand the discussion about the interpretation of their theoretical findings, in particular explaining in more detail their computational and privacy arguments in the context of the full exponential family?
>
> In Appendix A.4, we expand the derivation of the theorectical findings. Please refer to that part.

---

> ### Comment · Reviewer_gCmo · 2023-11-22
> **A quick response, before reading the rest**
>
> Note that neither me, nor the other reviewers seem to see your updated paper. Can the authors upload it?
> Edit: Now seem to be ok. Thank you.

---

> ### Author Response · Authors · 2023-11-22
> **Response to Reviewer gCmo (Part 2)**
>
> > Q4: Can the authors expand the focal loss experiments to more settings, especially ones where the success rate is not 100%?
>
> Here, we present additional experiments conducted on the Focal Loss. We mainly test the parameters of $\tau$, $\gamma$ and $\varepsilon$ on an untrained model, and average the experiments over 10 trials. Focusing on temperature $\tau$, we present several cases where the accuracy is not 100%. In addition, by varying $\gamma$ and $\varepsilon$ on these setting, the accuracies are not affected.
>
> Temperature $\tau$ (batch size=64, activation=ReLU):
>
> | Dataset   | Model     | $\tau$ | Our ClsAcc | Our InsAcc |
> | --------- | --------- | ------ | ---------- | ---------- |
> | MNIST     | LeNet     | 0.5    | 0.980      | 0.906      |
> |           |           | 0.75   | 0.980      | 0.945      |
> |           |           | 0.9    | 0.990      | 0.954      |
> |           |           | 1.25   | 0.990      | 0.983      |
> |           |           | 1.5    | 1.000      | 0.998      |
> | CIFAR-10  | ResNet-18 | 0.5    | 0.990      | 0.960      |
> |           |           | 0.75   | 1.000      | 0.994      |
> |           |           | 0.9    | 1.000      | 1.000      |
> |           |           | 1.25   | 1.000      | 1.000      |
> |           |           | 1.5    | 1.000      | 1.000      |
> | CIFAR-100 | ResNet-50 | 0.5    | 1.000      | 1.000      |
> |           |           | 0.75   | 1.000      | 1.000      |
> |           |           | 0.9    | 1.000      | 1.000      |
> |           |           | 1.25   | 1.000      | 1.000      |
> |           |           | 1.5    | 1.000      | 1.000      |
>
> We have observed that the temperature parameter, $\tau$, significantly impacts smaller datasets. When $\tau$ is smaller, the space of logits expands, complicating the estimation of batch posterior probabilities. Consequently, as $\tau$ decreases, label accuracy deteriorates. For the large datasets with more classes, such as CIFAR-10, the logits space is hardly influenced by changing different $\tau$.
>
> > Q5: Can the authors provide a discussion about why the focal loss is significant in particular? Are there other common classification losses that it covers as sub-cases? Are there other common classification losses not covered by the focal loss that might be more secure?
>
> - To the best of our knowledge, Focal Loss has the general form in the cross-entropy loss variants and it can be converted to CE loss or BCE loss by setting different $\alpha$ and $\gamma$. We aim to derive a general form of label leakage from gradient, so we choose the Focal Loss.
> - In supervised learning, we currently did not find other common classification losses that can cover the Focal Loss.
> - There are indeed some other classification losses not covered by the Focal Loss, such as the the Hinge Loss used in Support Vector Machines or the Kullback-Leibler Divergence used in some probabilistic models. However, this paper cannot conclude they are more or less secure than the Focal Loss. Because this paper only covers the classification loss function with cross-entropy and softmax (or sigmoid).

---

> ### Author Response · Authors · 2023-11-22
> **Response to Reviewer gCmo (Part 3)**
>
> > Q8: Can the authors provide a comparison with [3] or [5] when auxiliary data is used for estimating their parameters?
>
> We name ZLG for [3] and LLG for [5]. The following experiments compare the label recovery accuracies with different model architectures, batch sizes and activation functions.
>
> - Batch size=64, activation=ReLU:
>
>   | Dataset   | Model     | ZLG ClsAcc | ZLG InsAcc | LLG ClsAcc | LLG InsAcc | Our ClsAcc | Our InsAcc |
>   | --------- | --------- | ---------- | ---------- | ---------- | ---------- | ---------- | ---------- |
>   | MNIST     | LeNet     | **1.000**  | 0.996      | **1.000**  | **1.000**  | 0.995      | 0.955      |
>   | CIFAR-10  | ResNet-18 | **1.000**  | 0.916      | 1.000      | 0.914      | **1.000**  | **1.000**  |
>   | CIFAR-100 | ResNet-50 | 0.985      | 0.893      | 0.816      | 0.633      | **1.000**  | **1.000**  |
>
> - Batch size=256, activation=ReLU:
>
>   | Dataset   | Model     | ZLG ClsAcc | ZLG InsAcc | LLG ClsAcc | LLG InsAcc | Our ClsAcc | Our InsAcc |
>   | --------- | --------- | ---------- | ---------- | ---------- | ---------- | ---------- | ---------- |
>   | MNIST     | LeNet     | **1.000**  | 0.977      | **1.000**  | **0.982**  | **1.000**  | 0.951      |
>   | CIFAR-10  | ResNet-18 | **1.000**  | 0.905      | **1.000**  | 0.919      | **1.000**  | **0.977**  |
>   | CIFAR-100 | ResNet-50 | 0.998      | 0.881      | **1.000**  | 0.868      | **1.000**  | **1.000**  |
>
> - Batch size=64, activation=ELU:
>
>   | Dataset   | Model     | ZLG ClsAcc | ZLG InsAcc | LLG ClsAcc | LLG InsAcc | Our ClsAcc | Our InsAcc |
>   | --------- | --------- | ---------- | ---------- | ---------- | ---------- | ---------- | ---------- |
>   | MNIST     | LeNet     | **1.000**  | 0.948      | 0.580      | 0.283      | 0.990      | **0.969**  |
>   | CIFAR-10  | ResNet-18 | **1.000**  | 0.902      | 0.980      | 0.697      | **1.000**  | **0.972**  |
>   | CIFAR-100 | ResNet-50 | 0.985      | 0.897      | 0.845      | 0.603      | **1.000**  | **1.000**  |
>
> > Q9: Can the authors test their methods in a FedAvg setting (check [3,6,7])?
>
> We present a concept for a label attack within the Federated Averaging (FedAvg) setting. Drawing inspiration from [7], we leverage a surrogate model to estimate the unshared gradients, denoted as $G_t$, during local training with FedAvg. At each iteration $t$, we update the model $M_t$ by considering the previous estimated state $M_{t-1}$, the update $G_{t-1}$, and the learning rate $\eta$. This update is represented by the equation $M_{t} = M_{t-1} - \eta \cdot G_{t-1}$. Utilizing the updated model $M_{t}$ and the estimated gradients $G_{t}$, we can calculate positive ($\hat p_j^+$) and negative ($\hat p_j^-$) probabilities for each class $j$. These probabilities are derived through our formulated expression for FedSGD algorithm. At this point, we can restore all the labels during the local updates.
>
> [7] Zhu, Junyi, Ruicong Yao, and Matthew B. Blaschko. "Surrogate model extension (SME): A fast and accurate weight update attack on federated learning." arXiv preprint arXiv:2306.00127 (2023).
>
> > Q10: Do the authors know why the SELU activation results in so much more variance in the estimates of its posterior probability distribution?
>
> We address this phenomenon by examining the output range $a$ of various activation functions. The input is denoted as $x$, and we all use the default settings in PyTorch.
>
> - Sigmoid: $a\in(0,1)$
> - Tanh: $a\in(-1,1)$
> - ReLU: $a\in[0, x]$
> - ELU: $a\in(-1,x]$
> - SELU: $a\in(-1.76,x]$
>
> Observing these output ranges, it becomes evident that SELU has the widest output space among these activation functions. Consequently, the output of each layer in the network may undergo larger shifts compared to the others. This increased shift results in a posterior probability distribution with higher variance, posing a greater challenge for attackers attempting estimation. Understanding this aspect sheds could explain why the attacks under SELU activation have a relative worse performance compared to the other activation functions.

---

> ### Comment · Reviewer_gCmo · 2023-11-22
> **Response to Response to Reviewer gCmo**
>
> In the interest of providing the authors with quick feedback before the end of the discussion period. I have skimmed their responses and the new pdf without going through it in detail yet. My feedback follows:
>
> **Response to Q1:** I am thankful to the authors. This indeed drastically helps the readability of the paper. While I haven't yet looked through all derivations from scratch, one thing I want to point out/ask about is interpreting $p_t$ as the softmax at the start of the proof on page 13. I thought, it could also be one minus the softmax instead, depending on the ground truth label. Can the authors elaborate on this/fix the derivations?
>
> **Response to Q2:** I am sure the authors should be able to find an intuitive interpretation within the exponential family for the focal loss ( given its closeness to CE ) and equal vulnerability. A quick google search suggested the following article [1]. Again, due to the interest of time, I haven't gone through it in detail to see if it exactly relevant to the author's use case.
>
> **Response to Q3:** In Appendix A.4. the authors discuss the computational savings of the gradient function of the softmax+CE combo. Are those savings realized by the backward pass of, say, pytorch? Are the savings significant for the training time? Can the authors give an example of another exponential family member's final gradient and how it does not have the same computation savings (both in terms of the formula and computation complexity)?
>
> [1] https://towardsdatascience.com/cross-entropy-classification-losses-no-math-few-stories-lots-of-intuition-d56f8c7f06b0

---

> > ### Comment · Reviewer_gCmo · 2023-11-22
> > **Response to Response to Reviewer gCmo**
> >
> > **Response to Q4:** I am thankful to the authors for providing this experiment. I am more confident now that the attack on the Focal Loss is as effective as the one on CE.
> >
> > **Response to Q8:** I assume all three methods use the same auxiliary data in this experiment? If so, I would argue that the shown results are very good and the **other reviewers** should also take a look at them, as they constitute one of the most important results of the paper.
> >
> > That said, I still suggest the authors, before the end of the discussion period, to write a paragraph pointing out where the big difference between them and prior work comes from.

---

> > > ### Author Response · Authors · 2023-11-23
> > > **Response to Reviewer gCmo (Part 4)**
> > >
> > > Thank you for your feedback and valuable suggestions.
> > >
> > > > That said, I still suggest the authors, before the end of the discussion period, to write a paragraph pointing out where the big difference between them and prior work comes from.
> > >
> > > Here is the explanation of the results from the tables.
> > >
> > > - For the LeNet model with ReLU activation, when the batch size increases from 64 to 256, all the attacks have a slight decrease in InsAcc. It is straightforward to understand that a larger batch size is much more difficult to attack than a smaller batch size.
> > >
> > > - For the CIFAR-10 dataset trained on LeNet or ResNet-18, we can observe that the InsAcc of ZLG and LLG is lower than our attack. When changing the shallow LeNet model to the deep ResNet-18 model, the InsAcc of ZLG and LLG decreases significantly. However, the InsAcc of our attack is improved, which indicates that our attack is more robust than ZLG and LLG on deep neural networks.
> > >
> > > -  For the CIFAR-100 dataset, it is shown that our attack reaches 100\% InsAcc on all the settings. We conclude that a dataset with more classes makes it easy for our attack to recover the labels. This is because the more classes there are, the estimated probabilities are more accurate and concentrated around the ground-truth value.
> > >
> > > - Finally, when changing the activation function from ReLU to ELU, the InsAcc of LLG drops dramatically. This is because LLG is based on the assumption that the activation function is non-negative. When the activation function is changed to ELU, the assumption is violated, and the InsAcc of LLG drops significantly.
> > >
> > > Moreover, we had add one row of comparison (CIFAR-10 on LeNet) for better illustration, which can be seen from Appendix D.2.
> > >
> > > ---
> > >
> > > According to your suggestions, we have made some modifications in the newly updated PDF paper.
> > >
> > > - In Appendix A, we add the ethics statement of this research.
> > >
> > > - In Appendix B, we make a comprehensive review of the label attacks (iDLG, GI, RLG, ZLG, LLG, iLRG) that are most related to this paer. We hope to discuss what is the difference between our attack and the previous studies.
> > >
> > > - In Appendix C, we add the definition of positive and negative samples, positive and negative probability, and the class-wise labels.
> > >
> > > - In Appendix D, we update the results on focal loss, comparison with ZLG and LLG, and the distribution shift of auxiliary dataset.
> > >
> > > Please let us know if you have any other questions or concerns. Thank you very much.

---

> ### Comment · Reviewer_gCmo · 2023-11-23
> **Response**
>
> I really appreciate Appendix B. It really helps me to evaluate the paper's contribution better.
>
> For the next revision, I will suggest authors provide a version of Table 5-7 including the main results -e.g iLRG and the version of LLG/ZLG that does not use auxiliary data (but dummy data), as I think it will benefit the community as a whole, as these works have been inconsistent at comparing and citing related work. I also think trying the author's approach on dummy data might be interesting.
>
> For the next revision, I will also encourage the authors to decrease the prominence of the exponential family discussion in the main paper and/or to extend it to the focal loss.
>
> I am very thankful to the authors for their rebuttal engagement as a whole, and I have raised my grade to 6. I might raise it further after going through the paper fully with the new information supplied. I could not do this in the short time frame given.

---

### Official Review · Reviewer_zaUc · 2023-10-31

**Soundness:** 3 good
**Presentation:** 2 fair
**Contribution:** 3 good
**Rating:** 5
**Confidence:** 4

**Summary:**

This paper proposed a label recovery attack by estimating the posterior probabilities. The authors first obtain the relationship between the gradient of focal loss and posterior probability. Then, they explain the essential reasons for such findings from the perspective of the exponential family. They also empirically observe that positive (negative) samples of a class have approximate probability distributions. Experiments on different datasets validate the effectiveness of the proposed method.

**Strengths:**

* The proposed method is well supported by theoretical analysis.
* The proposed method empirically works.

**Weaknesses:**

* The proposed method requires an auxiliary dataset with the same distribution of training data. This requirement is impractical in FL.
* The writing is not clear. For example, I cannot find the exact definition of 1) negative probabilities and positive probabilities in Sec 5.1, class-wise labels in Theorem 2.
* [Minor] The second equation in the proof for theorem 3, Appendix A.1 should be

$$
\nabla_{z_j}L_{FL}
=\sum_{t=1}^K\alpha_t\frac{\partial\bar{h}}{\partial z_j}\log\sum_{k=1}^Ke^{z_k}+\sum_{t=1}^K\alpha_t(1-p_t)^{\gamma}p_j-\sum_{t=1}^K\alpha_t\frac{\partial\bar{h}}{\partial z_j}\textcolor{red}{z_t}-\alpha_j(1-p_j)^\gamma.
$$

$z_t$ is missing in the equation.

**Questions:**

The performance of the proposed attack in Theorem 2 relies on the quality of posterior probability. To my understanding, The posterior probability increases when the training epoch increases. Therefore, the performance should increase with the training epoch. However, in Figure 4, the performance decreases as the training epoch increases. Could you please provide more explanation?

---

> ### Author Response · Authors · 2023-11-20
> **Response to Reviewer zaUc**
>
> Thanks for appreciation of our theorectical analysis and empirical evaluations, and pointing out the imperceptible mistakes. We give a point-by-point response to the weaknesses and questions, and hope to clear up your concerns.
>
> > Weakness 1: The proposed method requires an auxiliary dataset with the same distribution of training data. This requirement is impractical in FL.
>
> We would like to highlight the following points to justify this assumption:
>
> 1. **Realistic Threat Model:** In many practical scenarios, attackers often have access to auxiliary data sources that closely resemble the distribution of the target training data. This assumption aligns with the notion that attackers may gather information from publicly available sources or other related datasets to enhance the effectiveness of their attacks.
> 2. **Common Practice in Security Analysis:** Making use of auxiliary data with a similar distribution is a standard practice in security analysis, especially in the research fields of *Membership Inference Attacks* [1, 2] and *Gradient Inversion Attacks* [3, 4]. This enables a more thorough evaluation of the model's robustness, covering a broader range of potential adversary scenarios.
> 3. **Worst-Case Scenario Exploration:** Assuming auxiliary data with a matching distribution, our analysis explores a worst-case scenario, where the adversary accesses highly relevant information. This reveals insights into the FL's vulnerabilities under challenging conditions, providing a comprehensive evaluation of its security properties.
>
> [1] Hu, Hongsheng, et al. "Membership inference attacks on machine learning: A survey." *ACM Computing Surveys (CSUR)* 54.11s (2022): 1-37.
>
> [2] Carlini, Nicholas, et al. "Membership inference attacks from first principles." *2022 IEEE Symposium on Security and Privacy (SP)*. IEEE, 2022.
>
> [3] Jeon, Jinwoo, et al. "Gradient inversion with generative image prior." *Advances in neural information processing systems* 34 (2021): 29898-29908.
>
> [4] Balunovic, Mislav, et al. "Lamp: Extracting text from gradients with language model priors." *Advances in Neural Information Processing Systems* 35 (2022): 7641-7654.
>
> > Weakness 2: I cannot find the exact definition of negative probabilities and positive probabilities in Sec 5.1, class-wise labels in Theorem 2.
>
> Here, we add the definition and explanation of the positive and negative probabilities, and the class-wise labels.
>
> In a multi-class classification problem, each instance in the dataset belongs to one of several classes. Let's denote the set of classes as $K$ and a particular class of interest as $k\in K$. In this context, we can define positive and negative samples for class $k$.
>
> 1. **Positive Samples** ($X_{k}^{+}$): The positive samples of class $k$ satisfy that: $X_{k}^{+}=\{x_i|y_i=k\}$, where $x_i$ is the input and $y_i$ is the corresponding label.
> 2. **Negative Samples** ($X_{k}^{-}$): Similarly, the negative samples of class $k$ satisfy that: $X_{k}^{-}=\{x_i|y_i\neq k\}$.
>
> According to the positive and negative samples, we can then get the positive and negative probability for class $k$.
>
> 1. **Positive Probability** ($p_{k}^{\text{+}}$): When a positive instance is fed into the model, the predicted probability of class  $k$ is termed the positive probability. Since the Softmax activation function is used in the output layer, the output posterior probability $p^{+}$ is a vector of length $k$. Therefore, the positive probability for class $k$ can be expressed as $p_{k}^{+}$.
> 2. **Negative Probability** ($p_{k}^{-}$): Similarly, when a negative sample is input into the model, the $k$th element of the output probability vector represents the negative probability, denoted as $p_{k}^{-}$. It's essential to note that any negative sample associated with the other $(K-1)$ classes contributes to $p_{k}^{-}$.
>
> When using an auxiliary dataset to esimate the probabilities of the target training batch in FL, we denote the estimated positive and negative probabilities as $\hat p_{k}^{+}$ and  $\hat p_{k}^{-}$, respectively.
>
> In a batch size of $B$, we aim to recover the labels of each instance within the batch, i.e., $\mathbf{y}=[y^{(1)},y^{(2)},\cdots,y^{(B)}]$. As this is a multi-class classification problem, the ground-truth labels $\mathbf{y}$ can also be represented the occurences of each class: $\mathbf{y}=[n_1,n_2,\cdots,n_K]$, where $n_k$ is the number of samples belonging to class $k$ and $K$ is the number of total classes.
>
> - **Class-wise Labels**: In a batch data, the class-wise labels can be defined as: $n_k=\sum_{i=1}^{B}\delta(y^{(i)}=k)$. Here, $n_k$ is the number of samples belonging to class $k$, B is the batch size, $y^{(i)}$ is the true class label of the $i$th instance in the batch, and $\delta(\cdot)$ is the Kronecker delta function, which equals 1 if the condition inside is true and 0 otherwise.

---

> > ### Author Response · Authors · 2023-11-20
> > **Response to Reviewer zaUc (Part 2)**
> >
> > > Weakness 3: The missing of $z_t$ in the second equation in the proof for theorem 3, Appendix A.1.
> >
> > Thank you for your careful observation and pointing out the mistake. We have fixed the mistake in the paper.
> >
> > > Q1: The performance of the proposed attack in Theorem 2 relies on the quality of posterior probability. To my understanding, The posterior probability increases when the training epoch increases. Therefore, the performance should increase with the training epoch. However, in Figure 4, the performance decreases as the training epoch increases. Could you please provide more explanation?
> >
> > - In the initial phase of model training, the model lacks the ability to differentiate between samples from each class. Consequently, the model assigns similar output probabilities to all fitted samples. It's important to note that, during this stage, the posterior probabilities for each class closely approximate $1/K$, with $K$ representing the number of classes in the current classification task.
> > - As training progresses, the model gradually learns distinctive features of each class. For the $i$th class, probabilities of positive samples (belonging to class $i$) increase, reflecting the model's growing confidence in its predictions. Conversely, negative probabilities (samples from class $j$ where $j\neq i$) decrease toward 0.
> > - However, varying learning speeds among different samples lead to a distribution shift in posterior probabilities during training. This results in a bias in estimating the batch data's posterior probability using samples from the auxiliary dataset. In summary, this explains the observed drop in InsAcc during model training.

---

> > > ### Comment · Reviewer_zaUc · 2023-11-22
> > > **Thank for the responses**
> > >
> > > Thanks to the author for the responses.
> > >
> > > The revision is not provided and the original manuscript is hard to understand due to unclear notation.
> > > The phenomenon that the learned posterior probabilities are worse than random guesses is quite confusing. Further justification or calibration techniques might be needed.
> > >
> > > Therefore, I would like to keep my rating.

---

> > > > ### Author Response · Authors · 2023-11-23
> > > > **Thanks for the feedback**
> > > >
> > > > Thank you for your feedback and for taking time to review our work. We apologize for any confusion caused by the unclear notation and understand your concerns regarding the learned posterior probabilities.
> > > >
> > > > We have already added the notions in the Appendix C to address these issues. We appreciate your patience and understanding, and we hope that our revised version will meet your expectations.
> > > >
> > > > Best regards,
> > > >
> > > > Authors of #4626

---

### Official Review · Reviewer_AjeW · 2023-11-05

**Soundness:** 3 good
**Presentation:** 3 good
**Contribution:** 3 good
**Rating:** 6
**Confidence:** 4

**Summary:**

The paper analyses root cause of label leakage from gradients and propose a novel attack which estimates the posterior probabilities from an auxiliary dataset.

**Strengths:**

1. Analysis of label leakage is novel and insightful, and the conclusion is interesting: combining cross-entropy loss and Softmax is intended to reduce computation but opens a backdoor to privacy attacks.
2. Novelty is clear.
3. Writing is easy to follow.

**Weaknesses:**

A small weakness: the attack assumes adversary can use an auxiliary data with the same distribution of training data, which in reality is not always true. I hope there can be some more discussions on how the results can be if auxiliary datasets and training datasets have considerable distribution shift.

**Questions:**

1. MNIST and CIFAR10 are relatively small and easy datasets. I wonder if the posterior probability estimation accuracy can scale up, if the data is more complex and the task is more challenging? For example, in ImageNet there are more vague and hard samples, the variance in the data distribution will be larger. Can you show us what will be the case in ImageNet or other large datasets?

2. In Figure 4 rightmost subfigure, why instance accuracy drops with model training?

---

> ### Author Response · Authors · 2023-11-20
> **Response to Reviewer AjeW**
>
> We appreciate your praise for the novelty and inspiration of our paper and some concerns about the assumption and experiments. We give point-by-point responses below. We are pleased to offer more explanation or revisions if you have any other recommendations.
>
> > Weakness 1: The attack assumes adversary can use an auxiliary data with the same distribution of training data, which in reality is not always true.
>
> We would like to highlight the following points to justify this assumption:
>
> 1. **Realistic Threat Model:** In many practical scenarios, attackers often have access to auxiliary data sources that closely resemble the distribution of the target training data. This assumption aligns with the notion that attackers may gather information from publicly available sources or other related datasets to enhance the effectiveness of their attacks.
> 2. **Common Practice in Security Analysis:** Making use of auxiliary data with a similar distribution is a standard practice in security analysis, especially in the research fields of *Membership Inference Attacks* [1] and *Gradient Inversion Attacks* [2]. This enables a more thorough evaluation of the model's robustness, covering a broader range of potential adversary scenarios.
> 3. **Worst-Case Scenario Exploration:** Assuming auxiliary data with a matching distribution, our analysis explores a worst-case scenario, where the adversary accesses highly relevant information. This reveals insights into the FL's vulnerabilities under challenging conditions, providing a comprehensive evaluation of its security properties.
>
> [1] Hu, Hongsheng, et al. "Membership inference attacks on machine learning: A survey." *ACM Computing Surveys (CSUR)* 54.11s (2022): 1-37.
>
> [2] Jeon, Jinwoo, et al. "Gradient inversion with generative image prior." *Advances in neural information processing systems* 34 (2021): 29898-29908.
>
> > Weakness 2: I hope there can be some more discussions on how the results can be if auxiliary datasets and training datasets have considerable distribution shift.
>
> To address concerns about significant distribution shifts, we conducted supplementary experiments, averaging results over 10 trials.
>
> 1. In the first set of experiments, we alternately used CIFAR-10 and CINIC-10 [3] as the training and auxiliary datasets. CINIC-10 extends CIFAR-10 by incorporating downsampled ImageNet images. While there is some overlap in the distributions due to similar object categories, there is a notable bias in the data features. We employed an untrained VGG model with a batch size of 64, and the results are presented in the table below.
>
>    | Training Data | Auxiliary Data | Model | Our ClsAcc | Our InsAcc |
>    | ------------- | -------------- | ----- | ---------- | ---------- |
>    | CIFAR-10      | CINIC-10       | VGG   | 1.000      | 1.000      |
>    | CINIC-10      | CIFAR-10       | VGG   | 1.000      | 1.000      |
>
> 2. Similarly, the second set of experiments involves alternating between MNIST and Fashion-MNIST as the training and auxiliary datasets. Despite both datasets having 10 classes, the objects they represent are entirely different. MNIST dataset contains a lot of handwritten digits, while Fashion-MNIST represents the article of clothing. Employing an untrained LeNet model with a batch size of 64, the results are presented in the table below.
>
>    | Training Data | Auxiliary Data | Model | Our ClsAcc | Our InsAcc |
>    | ------------- | -------------- | ----- | ---------- | ---------- |
>    | MNIST         | Fashion-MNIST  | LeNet | 1.000      | 0.969      |
>    | Fashion-MNIST | MNIST          | LeNet | 1.000      | 0.948      |
>
>
> This phenomenon can be explained as follows: In the initial phase of model training, the model lacks the ability to differentiate between samples from each class, assigning similar output probabilities to all fitted samples (i.e., $1/K$). For different training datasets like Fashion-MNIST and MNIST, the model projects input figures to similar output probabilities with slight variations. This benefits our label recovery attack as it becomes easier to estimate the posterior probabilities of the target batch.
>
> Given that Fashion-MNIST is more intricate than MNIST, utilizing MNIST as the auxiliary dataset poses challenges in accurately estimating the probability distribution of batch training samples. This difficulty accounts for the marginal decrease in InsAcc. However, since CIFAR-10 and CINIC-10 exhibit similarities, there is no difference in InsAcc. The outcomes of these experiments illustrate that if an attacker initiates an attack during the early training stages, having an auxiliary dataset with an identical distribution to the training dataset may not be essential.
>
> [3] Darlow, Luke N., et al. "Cinic-10 is not imagenet or cifar-10." *arXiv preprint arXiv:1810.03505* (2018).

---

> > ### Author Response · Authors · 2023-11-20
> > **Response to Reviewer AjeW (Part 2)**
> >
> > > Q1: If the posterior probability estimation accuracy can scale up, if the data is more complex and the task is more challenging?
> >
> > We conducted additional experiments on ImageNet, consistently achieving 100% class-wise accuracy (ClsAcc) and instance-wise accuracy (InsAcc). Moreover, our observations indicate that applying our attack to a more complex and larger dataset with more classes yields amazing results.
> >
> > Furthermore, we performed comparative experiments, investigating variations in model architecture, batch size, and activation function. All models tested were untrained, and we used ImageNet as the auxiliary dataset. The results of these experiments are presented in the table below.
> >
> > - Model architectures:
> >
> >   | Model     | Batch size | Activation | Our ClsAcc | Our InsAcc |
> >   | --------- | ---------- | ---------- | ---------- | ---------- |
> >   | ResNet-50 | 64         | ReLU       | 1.000      | 1.000      |
> >   | ResNet-34 | 64         | ReLU       | 1.000      | 1.000      |
> >   | VGG-11    | 64         | ReLU       | 1.000      | 1.000      |
> >
> > - Batch sizes:
> >
> >   | Model     | Batch size | Activation | Our ClsAcc | Our InsAcc |
> >   | --------- | ---------- | ---------- | ---------- | ---------- |
> >   | ResNet-50 | 64         | ReLU       | 1.000      | 1.000      |
> >   | ResNet-50 | 256        | ReLU       | 1.000      | 1.000      |
> >   | ResNet-50 | 512        | ReLU       | 1.000      | 1.000      |
> >
> > - Activation functions:
> >
> >   | Model     | Batch size | Activation | Our ClsAcc | Our InsAcc |
> >   | --------- | ---------- | ---------- | ---------- | ---------- |
> >   | ResNet-50 | 64         | LeakyReLU  | 1.000      | 1.000      |
> >   | ResNet-50 | 64         | ELU        | 1.000      | 1.000      |
> >   | ResNet-50 | 64         | SELU       | 1.000      | 1.000      |
> >
> > As the number of classes of the dataset increases, the model tends to output smaller probabilities for each class. For example, the untrained model on ImageNet produces a probability of nearly 0.001 for each class, no matter the samples within this class or not. Employing the auxiliary dataset with same distribution enables us to obtain accurate estimates of the batch probabilities, facilitating the precise recovery of the number of labels per class. Importantly, the estimated probability distribution remains consistent across different model architectures, activation functions, and batch sizes, highlighting the attack's robustness, especially on larger or more complex datasets.
> >
> >
> >
> > > Q2: In Figure 4 rightmost subfigure, why instance accuracy drops with model training?
> >
> > - In the initial phase of model training, the model lacks the ability to differentiate between samples from each class. Consequently, the model assigns similar output probabilities to all fitted samples. It's important to note that, during this stage, the posterior probabilities for each class closely approximate $1/K$, with $K$ representing the number of classes in the current classification task.
> > - As training progresses, the model gradually learns distinctive features of each class. For the $i$th class, probabilities of positive samples (belonging to class $i$) increase, reflecting the model's growing confidence in its predictions. Conversely, negative probabilities (samples from class $j$ where $j\neq i$) decrease toward 0.
> > - However, varying learning speeds among different samples lead to a distribution shift in posterior probabilities during training. This results in a bias in estimating the batch data's posterior probability using samples from the auxiliary dataset. In summary, this explains the observed drop in InsAcc during model training.

---

> > > ### Comment · Reviewer_AjeW · 2023-11-22
> > > **Thanks for the response**
> > >
> > > I would like to express my gratitude to the authors for providing a thorough response to my questions, which effectively addressed my concerns. My stance remains in favor of acceptance, and therefore, I will maintain my original score.

---

> > > > ### Author Response · Authors · 2023-11-23
> > > > **Thanks for the review**
> > > >
> > > > Thank you for your kind words and for taking the time to review our work. We appreciate your positive feedback and are glad to hear that our responses were able to address your concerns effectively. Your valuable comments benefit us a lot.
> > > >
> > > > Best Regards,
> > > > Authors of #4626

---

### Meta-Review · Area_Chair_Vo5N · 2023-12-11

**Metareview:**

This paper starts with analyzing the label leakage problem in Fl from the perspective of focal loss, which seems to be quite novel. Experiments are comprehensive to demonstrate the proposed attack. I would encourage the authors to take in account the review feedback to improve the paper.

Strength:
1. An novel perspective from focal loss to understand label leakage problem in FL.
2. Good experimental results.

Weakness:
1. The assumption of auxiliary dataset with similar distribution is not very explained. The authors did give some explanations, but this still seems to be very strong assumption in my opinion. As stated in the paper, this auxiliary dataset is a direct subsample of original data, which is too close to the original data. Is there a way to construct data that is not that close and see how this works?
2. There are still several concerns not addressed during rebuttal as stated in the review by gCmo.

**Justification For Why Not Higher Score:**

See weakness.

**Justification For Why Not Lower Score:**

N/A

---

### Decision · Program_Chairs · 2024-01-16

Reject